# Nearly Minimax Optimal Reinforcement Learning for Discounted MDPs

**Jiafan He**
Department of Computer Science
University of California, Los Angeles
CA 90095, USA
jiafanhe19@ucla.edu

**Dongruo Zhou**
Department of Computer Science
University of California, Los Angeles
CA 90095, USA
drzhou@cs.ucla.edu

**Quanquan Gu**
Department of Computer Science
University of California, Los Angeles
CA 90095, USA
qgu@cs.ucla.edu

## Abstract

We study the reinforcement learning problem for discounted Markov Decision Processes (MDPs) under the tabular setting. We propose a model-based algorithm named UCBVI-$\gamma$, which is based on the *optimism in the face of uncertainty principle* and the Bernstein-type bonus. We show that UCBVI-$\gamma$ achieves an $\widetilde{O}\big(\sqrt{SAT}/(1-\gamma)^{1.5}\big)$ regret, where $S$ is the number of states, $A$ is the number of actions, $\gamma$ is the discount factor and $T$ is the number of steps. In addition, we construct a class of hard MDPs and show that for any algorithm, the expected regret is at least $\widetilde{\Omega}\big(\sqrt{SAT}/(1-\gamma)^{1.5}\big)$. Our upper bound matches the minimax lower bound up to logarithmic factors, which suggests that UCBVI-$\gamma$ is nearly minimax optimal for discounted MDPs.

## 1 Introduction

The goal of reinforcement learning (RL) is designing algorithms to learn the optimal policy through interactions with the unknown dynamic environment. Markov decision process (MDPs) plays a central role in reinforcement learning due to their ability to describe the time-independent state transition property. More specifically, the discounted MDP is one of the standard MDPs in reinforcement learning to describe sequential tasks without interruption or restart. For discounted MDPs, with a *generative model* [12], several algorithms with near-optimal sample complexity have been proposed. More specifically, Azar et al. [3] proposed an Empirical QVI algorithm which achieves the optimal sample complexity to find the optimal value function. Sidford et al. [22] proposed a sublinear randomized value iteration algorithm that achieves a near-optimal sample complexity to find the optimal policy, and Sidford et al. [23] further improved it to reach the optimal sample complexity. Since generative model is a powerful oracle that allows the algorithm to query the reward function and the next state for any state-action pair $(s, a)$, it is natural to ask whether there exist online RL algorithms (without generative model) that achieve optimality.

To measure an online RL algorithm, a widely used notion is *regret*, which is defined as the summation of sub-optimality gaps over time steps. The regret is firstly introduced for episodic and infinite-horizon average-reward MDPs and later extended to discounted MDPs by [15, 30, 35, 35]. Liu and Su [15] proposed a double Q-learning algorithm with the UCB exploration (Double Q-learning),

which enjoys $\widetilde{O}(\sqrt{SAT}/(1-\gamma)^{2.5})$ regret, where $S$ is the number of states, $A$ is the number of actions, $\gamma$ is the discount factor and $T$ is the number of steps. While Double Q-learning enjoys a standard $\sqrt{T}$-regret, it still does not match the lower bound proved in [15] in terms of the dependence on $S, A$ and $1/(1-\gamma)$. Recently, Zhou et al. [34] proposed a UCLK$^+$ algorithm for discounted MDPs under the linear mixture MDP assumption and achieved $\widetilde{O}(d\sqrt{T}/(1-\gamma)^{1.5})$ regret, where $d$ is the dimension of the feature mapping. However, directly applying their algorithm to our setting would yield an $\widetilde{O}(S^2 A\sqrt{T}/(1-\gamma)^{1.5})$ regret[1], which is even worse that of double Q-learning [15] in terms of the dependence on $S, A$.

In this paper, we aim to close this gap by designing a practical algorithm with a nearly optimal regret. In particular, we propose a model-based algorithm named UCBVI-$\gamma$ for discounted MDPs without using the generative model. At the core of our algorithm is to use a "refined" Bernstein-type bonus and the *law of total variance* [3, 4], which together can provide tighter upper confidence bound (UCB). Our contributions are summarized as follows:

- We propose a model-based algorithm UCBVI-$\gamma$ to learn the optimal value function under the discounted MDP setting. We show that the regret of UCBVI-$\gamma$ in first $T$ steps is upper bounded by $\widetilde{O}(\sqrt{SAT}/(1-\gamma)^{1.5})$. Our regret bound strictly improves the best existing regret $\widetilde{O}(\sqrt{SAT}/(1-\gamma)^{2.5})$[2] in [15] by a factor of $(1-\gamma)^{-1}$.

- We also prove a lower bound of the regret by constructing a class of hard-to-learn discounted MDPs, which can be regarded as a *chain* of the hard MDPs considered in [15]. We show that for any algorithm, its regret in the first $T$ steps can not be lower than $\widetilde{\Omega}(\sqrt{SAT}/(1-\gamma)^{1.5})$ on the constructed MDP. This lower bound also strictly improves the lower bound $\Omega(\sqrt{SAT}/(1-\gamma) + \sqrt{AT}/(1-\gamma)^{1.5})$ proved by [15].

- The nearly matching upper and the lower bounds together suggest that the proposed UCBVI-$\gamma$ algorithm is minimax-optimal up to logarithmic factors.

We compare the regret of UCBVI-$\gamma$ with previous online algorithms for learning discounted MDPs in Table 1.

**Notation** For any positive integer $n$, we denote by $[n]$ the set $\{1, \ldots, n\}$. For any two numbers $a$ and $b$, we denote by $a \vee b$ as the shorthand for $\max(a, b)$. For two sequences $\{a_n\}$ and $\{b_n\}$, we write $a_n = O(b_n)$ if there exists an absolute constant $C$ such that $a_n \leq Cb_n$, and we write $a_n = \Omega(b_n)$ if there exists an absolute constant $C$ such that $a_n \geq Cb_n$. We use $\widetilde{O}(\cdot)$ and $\widetilde{\Omega}(\cdot)$ to further hide the logarithmic factors.

## 2 Related Work

**Model-free Algorithms for Discounted MDPs.** A large amount of reinforcement learning algorithms like Q-learning can be regarded as model-free algorithms. These algorithms directly learn the action-value function by updating the values of each state-action pair. Kearns and Singh [13] firstly proposed a phased Q-Learning which learns an $\epsilon$-optimal policy with $\widetilde{O}(SA/((1-\gamma)^7\epsilon^2))$ sample complexity for $\epsilon \leq 1/(1-\gamma)$. Later on, Strehl et al. [25] proposed a delay-Q-learning algorithm, which achieves $\widetilde{O}(SA/((1-\gamma)^8\epsilon^4))$ sample complexity of exploration. Wang [29] proposed a randomized primal-dual method algorithm, which improves the sample complexity to $\widetilde{O}(SA/((1-\gamma)^4\epsilon^2))$ for $\epsilon \leq 1/(1-\gamma)$ under the ergodicity assumption. Later, Sidford et al. [23] proposed a sublinear randomized value iteration algorithm and achieved $\widetilde{O}(SA/((1-\gamma)^4\epsilon^2))$ sample complexity for $\epsilon \leq 1$. Sidford et al. [22] further improved the empirical QVI algorithm and proposed a variance-reduced QVI algorithm, which improves the sample complexity to $\widetilde{O}(SA/((1-\gamma)^3\epsilon^2))$ for $\epsilon \leq 1$. Wainwright [28] proposed a variance-reduced Q-learning algorithm, which is an extension of the Q-learning algorithm and achieves $\widetilde{O}(SA/((1-\gamma)^3\epsilon^2))$ sample complexity. In addition, Dong

---

[1]Linear mixture MDP assumes that there exists a feature mapping $\phi(s'|s, a) \in \mathbb{R}^d$ and a vector $\boldsymbol{\theta} \in \mathbb{R}^d$ such that $\mathbb{P}(s'|s, a) = \langle \phi(s'|s, a), \boldsymbol{\theta} \rangle$. It can be verified that any MDP is automatically a linear mixture MDP with a $S^A$-dimensional feature mapping [2, 35].

[2]The regret definition in [15] differs from our definition by a factor of $(1-\gamma)^{-1}$. Here we translate their regret from their definition to our definition for a fair comparison. A detailed comparison can be found in Appendix.

Table 1: Comparison of RL algorithms for discounted MDPs in terms of sample complexity and regret. Note that the regret bounds for all the compared algorithms except Double Q-learning [15] are derived from their sample complexity results. See Appendix A.1 for more details.

| | Algorithm | Sample complexity | Regret |
|---|---|---|---|
| Model-free | Delay-Q-learning [25] | $\widetilde{O}\left(\frac{SA}{(1-\gamma)^8\epsilon^4}\right)$ | $\widetilde{O}\left(\frac{S^{1/5}A^{1/5}T^{4/5}}{(1-\gamma)^{9/5}}\right)$ |
| | Q-learning with UCB [9] | $\widetilde{O}\left(\frac{SA}{(1-\gamma)^7\epsilon^2}\right)$ | $\widetilde{O}\left(\frac{S^{1/3}A^{1/3}T^{2/3}}{(1-\gamma)^{8/3}}\right)$ |
| | UCB-multistage [33] | $\widetilde{O}\left(\frac{SA}{(1-\gamma)^{5.5}\epsilon^2}\right)$ | $\widetilde{O}\left(\frac{S^{1/3}A^{1/3}T^{2/3}}{(1-\gamma)^{13/6}}\right)$ |
| | UCB-multistage-adv [33] | $\widetilde{O}\left(\frac{SA}{(1-\gamma)^3\epsilon^2}\right)$[3] | $\widetilde{O}\left(\frac{S^{1/3}A^{1/3}T^{2/3}}{(1-\gamma)^{4/3}}\right)$ |
| | Double Q-learning [15] | N/A | $\widetilde{O}\left(\frac{\sqrt{SAT}}{(1-\gamma)^{2.5}}\right)$ |
| Model-based | R-max [5] | $\widetilde{O}\left(\frac{S^2A}{(1-\gamma)^6\epsilon^3}\right)$ | $\widetilde{O}\left(\frac{S^{1/2}A^{1/4}T^{3/4}}{(1-\gamma)^{7/4}}\right)$ |
| | MoRmax [27] | $\widetilde{O}\left(\frac{SA}{(1-\gamma)^6\epsilon^2}\right)$ | $\widetilde{O}\left(\frac{S^{1/3}A^{1/3}T^{2/3}}{(1-\gamma)^{7/3}}\right)$ |
| | UCRL [14] | $\widetilde{O}\left(\frac{S^2A}{(1-\gamma)^3\epsilon^2}\right)$ | $\widetilde{O}\left(\frac{S^{2/3}A^{1/3}T^{2/3}}{(1-\gamma)^{4/3}}\right)$ |
| | UCBVI-$\gamma$ (**Our work**) | N/A | $\widetilde{O}\left(\frac{\sqrt{SAT}}{(1-\gamma)^{1.5}}\right)$ |
| Lower bound | N/A | $\widetilde{\Omega}\left(\frac{SA}{(1-\gamma)^3\epsilon^2}\right)$ [14] | $\widetilde{\Omega}\left(\frac{\sqrt{SAT}}{(1-\gamma)^{1.5}}\right)$ (**Our work**) |

2. It holds when $\epsilon \le 1/\text{poly}(S, A, 1/(1-\gamma))$.

et al. [9] proposed an infinite Q-learning with UCB and improved the sample complexity of exploration to $\widetilde{O}(SA/((1-\gamma)^7\epsilon^2))$. Zhang et al. [33] proposed a UCB-multistage algorithm which attains the $\widetilde{O}(SA/((1-\gamma)^{5.5}\epsilon^2))$ sample complexity of exploration, and proposed a UCB-multistage-adv algorithm which attains a better sample complexity $\widetilde{O}(SA/((1-\gamma)^3\epsilon^2))$ in the high accuracy regime. Recently, Liu and Su [15] focused on regret minimization for the infinite-horizon discounted MDP and showed the connection between regret and sample complexity of exploration. Liu and Su [15] proposed a Double Q-Learning algorithm, which achieves $\widetilde{O}(\sqrt{SAT}/(1-\gamma)^{2.5})$ regret within $T$ steps. Furthermore, Liu and Su [15] constructed a series of hard MDPs and showed that the expected regret for any algorithm is lower bounder by $\widetilde{\Omega}(\sqrt{SAT}/(1-\gamma) + \sqrt{AT}/(1-\gamma)^{1.5})$. There still exists a $1/(1-\gamma)$-gap between the upper and lower regret bounds. In contrast to the aforementioned model-free algorithms, our proposed algorithm is model-based.

**Model-based Algorithms for Discounted MDP.** Our UCBVI-$\gamma$ falls into the category of model-based reinforcement learning algorithms. Model-based algorithms maintain a model of the environment and update it based on the observed data. They will form the policy based on the learnt model. More specifically, to learn the $\epsilon$-optimal value function, Azar et al. [3] proposed an empirical QVI algorithm which achieves $\widetilde{O}(SA/((1-\gamma)^3\epsilon^2))$ sample complexity. Azar et al. [3] proposed an empirical QVI algorithm which improves the sample complexity to $\widetilde{O}(SA/((1-\gamma)^3\epsilon^2))$ for $\epsilon \le 1/\sqrt{(1-\gamma)S}$. Szita and Szepesvári [27] proposed an MoRmax algorithm, which achieves $\widetilde{O}(SA/((1-\gamma)^6\epsilon^2))$ sample complexity. Later, Lattimore and Hutter [14] proposed a UCRL algorithm, which achieves $\widetilde{O}(S^2A/((1-\gamma)^3\epsilon^2))$ sample complexity in general and $\widetilde{O}(SA/((1-\gamma)^3\epsilon^2))$ sample complexity with a strong assumption on the state transition. Recently, Agarwal et al. [1] proposed a refined analysis for the empirical QVI algorithm which achieves $\widetilde{O}(SA/((1-\gamma)^3\epsilon^2))$ sample complexity when $\epsilon \le 1/\sqrt{1-\gamma}$.

**Upper and Lower Bounds for Episodic MDPs.** There is a line of work which aims at proving sample complexity or regret for episodic MDPs (MDPs which consist of restarting episodes) [7, 18,

4, 19, 11, 8, 24, 21, 31, 32, 17, 20]. Compared with the episodic MDP, discounted MDPs involve only one infinite-horizon sample trajectory, suggesting that any two states or actions on the trajectory are dependent. Such a dependence makes the learning of discounted MDPs more challenging.

## 3 Preliminaries

We consider infinite-horizon discounted Markov Decision Processes (MDP) which are defined by a tuple $(\mathcal{S}, \mathcal{A}, \gamma, r, \mathbb{P})$. Here $\mathcal{S}$ is the state space with $|\mathcal{S}| = S$, $\mathcal{A}$ is the action space with $|\mathcal{A}| = A$, $\gamma \in (0, 1)$ is the discount factor, $r : \mathcal{S} \times \mathcal{A} \to [0, 1]$ is the reward function, $\mathbb{P}(s'|s, a)$ is the transition probability function, which denotes the probability that state $s$ transfers to state $s'$ with action $a$. For simplicity, we assume the reward function is *deterministic and known*. A *non-stationary policies* $\pi$ is a collection of function $\{\pi_t\}_{t=1}^{\infty}$, where each function $\pi_t : \{\mathcal{S} \times \mathcal{A}\}^{t-1} \times \mathcal{S} \to \mathcal{A}$ maps history $\{s_1, a_1, ..., s_{t-1}, a_{t-1}, s_t = s\}$ to an action. For any non-stationary policy $\pi$, we denote $\pi_t(s) = \pi_t(s; s_1, a_1, ..., s_{t-1}, a_{t-1})$ for simplicity. We define the action-value function and value function at step $t$ as follows:

$$Q_t^\pi(s, a) = \mathbb{E}\left[ \sum_{i=0}^{\infty} \gamma^i r(s_{t+i}, a_{t+i}) \Big| s_1, ..., s_t = s, a_t = a \right],$$

$$V_t^\pi(s) = \mathbb{E}\left[ \sum_{i=0}^{\infty} \gamma^i r(s_{t+i}, a_{t+i}) \Big| s_1, ..., s_t = s \right],$$

where $a_{t+i} = \pi_{t+i}(s_{t+i})$, and $s_{t+i+1} \sim \mathbb{P}\big( \cdot |s_{t+i}, \pi_{t+i}(s_{t+i})\big)$. In addition, we denote the optimal action-value function and the optimal value function as $Q^*(s, a) = \sup_\pi Q_t^\pi(s, a)$ and $V^*(s) = \sup_\pi V_1^\pi(s)$ respectively. Note that the optimal action-value function and the optimal value function are independent of the step $t$. For simplicity, for any function $V : \mathcal{S} \to R$, we denote $[\mathbb{P}V](s, a) = \mathbb{E}_{s' \sim \mathbb{P}(\cdot|s,a)}V(s')$. According to the definition of the value function, we have the following non-stationary Bellman equation and Bellman optimality equation for non-stationary policy $\pi$ and optimal policy $\pi^*$:

$$Q_t^\pi(s, a) = r(s, a) + \gamma[\mathbb{P}V_{t+1}^\pi](s, a), \ Q^*(s, a) = r(s, a) + \gamma[\mathbb{P}V^*](s, a). \tag{3.1}$$

## 4 Main Results

### 4.1 Algorithm

In this subsection, we propose the Upper Confidence Bound Value Iteration-$\gamma$ (UCBVI-$\gamma$) algorithm, which is illustrated in Algorithm 1. The algorithm framework of UCBVI-$\gamma$ follows the UCBVI algorithm proposed in Azar et al. [4], which can be regarded as the counterpart of UCBVI-$\gamma$ in the episodic MDP setting.

UCBVI-$\gamma$ is a model-based algorithm that maintains an empirical measure $\mathbb{P}_t$ at each step $t$. At the beginning of the $t$-th iteration, UCBVI-$\gamma$ takes action $a_t$ based on the greedy policy induced by $Q_t(s_t, a)$ and transits to the next state $s_{t+1}$. After receiving the next state $s_{t+1}$, UCBVI-$\gamma$ computes the empirical transition probability function $\mathbb{P}_t(s'|s, a)$ in (4.1). Based on empirical transition probability function $\mathbb{P}_t(s'|s, a)$, UCBVI-$\gamma$ updates $Q_{t+1}(s, a)$ by performing one-step value iteration on $Q_t(s, a)$ with an additional upper confidence bound (UCB) term $\text{UCB}_t(s, a)$ defined in (4.3). Here the UCB bonus term is used to measure the uncertainty of the expectation of the value function $V_t(s)$. Unlike previous work, which adapts a Hoeffding-type bonus [15], our UCBVI-$\gamma$ uses a Bernstein-type bonus which brings a tighter upper bound by accessing the variance of $V_t(s)$, denoted by $\text{Var}_{s' \sim \mathbb{P}(\cdot|,s,a)}V_t(s')$. However, since the probability transition $\mathbb{P}(\cdot|s, a)$ is unknown, it is impossible to calculate the exact variance of $V_t$. Instead, UCBVI-$\gamma$ estimates the variance by considering the variance of $V_t$ over the empirical probability transition function $\mathbb{P}_t(\cdot|s, a)$ defined in (4.1). Therefore, the final UCB bonus term in (4.3) can be regarded as a standard Bernstein-type bonus on the empirical measure $\mathbb{P}_t(\cdot|s, a)$ with an additional error term.

Compared with UCBVI algorithm in Azar et al. [4], the action-value function $Q_t(s, a)$ in UCBVI-$\gamma$ is updated in a forward way from step 1 to step $T$ with the initial value $Q_1(s, a) = 1/(1 - \gamma)$ for all $s \in \mathcal{S}, a \in \mathcal{A}$, while UCBVI updates its action-value function in a backward way from $Q_{t,H}$ to $Q_{t,1}$ with initial value $Q_{t,H}(s, a) = 0$. Compared with UCRL in Lattimore and Hutter [14], UCBVI-$\gamma$ does not need to call an additional extended value iteration sub-procedure [10, 26], which is not easy to implement even with infinite computation [14].

---

**Algorithm 1** Upper Confidence Value-iteration UCBVI-$\gamma$

---

1: Receive state $s_1$ and set initial value function $Q_1(s,a) \leftarrow 1/(1-\gamma)$, $N_0(s,a) = N_0(s,a,s') = N_0(s) \leftarrow 0$ for all $s \in \mathcal{S}, a \in \mathcal{A}, s' \in \mathcal{S}$
2: **for** step $t = 1, \ldots$ **do**
3:      Let $\pi_t(\cdot) \leftarrow \text{argmax}_{a \in \mathcal{A}} Q_t(\cdot, a)$, take action $a_t \leftarrow \pi_t(s_t)$ and receive next state $s_{t+1} \sim \mathbb{P}(\cdot|s_t, a_t)$
4:      Set $N_t(s) \leftarrow N_{t-1}(s)$, $N_t(s,a) \leftarrow N_{t-1}(s,a)$ and $N_t(s,a,s') \leftarrow N_{t-1}(s,a,s')$ for all $s \in \mathcal{S}, a \in \mathcal{A}, s' \in \mathcal{S}$
5:      Update $N_t(s_t) \leftarrow N_t(s_t) + 1$, $N_t(s_t, a_t) \leftarrow N_t(s_t, a_t) + 1$ and $N_t(s_t, a_t, s_{t+1}) \leftarrow N_t(s_t, a_t, s_{t+1}) + 1$
6:      For all $s \in \mathcal{S}, a \in \mathcal{A}$, set

$$\mathbb{P}_t(s'|s,a) = \frac{N_t(s,a,s')}{N_t(s,a) \vee 1}. \tag{4.1}$$

7:      Update new value function $Q_{t+1}(s,a)$ and $V_{t+1}(s)$ by

$$Q_{t+1}(s,a) = \min\left\{Q_t(s,a), r(s,a) + \gamma[\mathbb{P}_t V_t](s,a) + \mathbb{C}\gamma\text{UCB}_t(s,a)\right\},$$
$$V_{t+1}(s) = \max_{a \in \mathcal{A}} Q_{t+1}(s,a). \tag{4.2}$$

     where

$$\text{UCB}_t(s,a) = \sqrt{\frac{8U\text{Var}_{s' \sim \mathbb{P}_t(\cdot|s,a)}(V_t(s'))}{N_t(s,a) \vee 1}} + \frac{8U/(1-\gamma)}{N_t(s,a) \vee 1}$$
$$+ \sqrt{\frac{8\sum_{s'} \mathbb{P}_t(s'|s,a)\min\left\{100B_t(s'), 1/(1-\gamma)^2\right\}}{N_t(s,a) \vee 1}}, \tag{4.3}$$

     and $B_t(s') = \beta / \left[(1-\gamma)^5\left(N_t(s') \vee 1\right)\right]$.
8: **end for**

---

**Computational complexity** In each step $t$, Algorithm 1 needs to first compute the empirical transition $\mathbb{P}_t$ and update the value function $V_{t+1}$ by one-step value iteration, which will cost $O(S^2A)$ time complexity for each update. However, the number of updates can be reduced by using the "batch" update scheme adapted in [10, 7] and in this case Algorithm 1 only needs to update the value function $V_{t+1}$ when the number of visits $N_t(s,a)$ doubles. With this update scheme, the number of updates is upper bounded by $O(SA\log T)$ and the total cost for updating the value function is $O(S^3A^2\log T)$. In addition, the Algorithm 1 still needs to choose the action with respect to the value function $V_t$ and it costs $O(AT)$ time complexity. Thus, the total computation complexity of the "batch" version of Algorithm 1 is $O(AT + S^3A^2\log T)$.

### 4.2 Regret Analysis

In this subsection, we provide the regret bound of UCBVI-$\gamma$. We first give the formal definition of the regret for the discounted MDP setting.

**Definition 4.1.** For a given non-stationary policy $\pi$, we define the regret Regret$(T)$ as follow:

$$\text{Regret}(T) = \sum_{t=1}^{T} \left[V^*(s_t) - V_t^\pi(s_t)\right].$$

The same regret has been used in prior work [30, 35, 34] on discounted MDPs. It is related to the "sample complexity of exploration" [12, 14, 9]. For more details about the connection between the regret and the sample complexity, please refer to Appendix A.

**Remark 4.2.** Without the use of generative model [12], an agent may enter bad states at the first few steps in discounted MDPs and there is no "restarting" mechanism as in episodic MDPs that can prevent the agent from being stuck in those bad states. Due to this limitation, both the regret and the sample complexity of exploration guarantees are not sufficient to ensure a good policy being learned. We think this is the fundamental limitation in the online learning of discounted MDPs.

With Definition 4.1, we introduce our main theorem, which gives an upper bound on the regret for UCBVI-$\gamma$.

**Theorem 4.3.** Let $U = \log(40SAT^3 \log^2 T/(\delta(1-\gamma)^2))$. If we set $\beta = S^2 A^2 U^5$ in UCBVI-$\gamma$, then with probability at least $1 - \delta$, the regret of UCBVI-$\gamma$ in Algorithm 1 is bounded by

$$\text{Regret}(T) \leq \frac{752 S^2 A^{1.5} U^{3.5}}{(1-\gamma)^{3.5}} + \frac{60 U \sqrt{SAT}}{(1-\gamma)^{1.5}} + \frac{4\sqrt{TU}}{(1-\gamma)^2}.$$

**Remark 4.4.** Notice that when $T = \widetilde{\Omega}(S^3 A^2/(1-\gamma)^4)$ and $SA = \Omega(1/(1-\gamma))$, the regret is bounded by $\widetilde{O}\big(\sqrt{SAT}/(1-\gamma)^{1.5}\big)$. In addition, since $\text{Regret}(T) \leq T/(1-\gamma)$ holds for any $T$, we have $\mathbb{E}[\text{Regret}(T)] = \widetilde{O}\big(\sqrt{SAT}/(1-\gamma)^{1.5} + T\delta/1-\gamma\big)$. When choosing $\delta = 1/T$, we have $\mathbb{E}[\text{Regret}(T)] = \widetilde{O}\big(\sqrt{SAT}/(1-\gamma)^{1.5}\big)$.

We also provide a regret lower bound, which suggests that our UCBVI-$\gamma$ is nearly minimax optimal.

**Theorem 4.5.** Suppose $\gamma \geq 2/3$, $A \geq 30$ and $T \geq 100SAL/(1-\gamma)^4$, then for any algorithm, there exists an MDP such that

$$\mathbb{E}[\text{Regret}(T)] \geq \frac{\sqrt{SAT}}{10000(1-\gamma)^{1.5}} - \frac{4\sqrt{STL}}{(1-\gamma)^{1.5}} - \frac{8S}{(1-\gamma)^2},$$

where $L = \log\big(300S^4 T^2/(1-\gamma)\big) \log(10ST)$.

**Remark 4.6.** When $T$ is large enough and $A = \widetilde{\Omega}(1)$, Theorem 4.5 suggests that the lower bound of regret is $\widetilde{\Omega}(\sqrt{SAT}/(1-\gamma)^{1.5})$. It can be seen that the regret of UCBVI-$\gamma$ in Theorem 4.3 matches this lower bound up to logarithmic factors. Therefore, UCBVI-$\gamma$ is nearly minimax optimal.

## 5 Proof of the Main Results

In this section, we provide the proofs of Theorems 4.3 and 4.5. The missing proofs are deferred to the appendix.

### 5.1 Proof of Theorem 4.3

In this subsection, we prove Theorem 4.3. For simplicity, let $\delta' = (1-\gamma)^2 \delta/(80T \log^2 T)$, then $U = \log(SAT^2/\delta')$. We first present the following key lemma, which shows that the optimal value functions $V^*$ and $Q^*$ can be upper bounded by the estimated functions $V_t$ and $Q_t$ with high probability:

**Lemma 5.1.** With probability at least $1 - 64T\delta \log^2 T/(1-\gamma)^2$, for all $t \in [T], s \in \mathcal{S}, a \in \mathcal{A}$, we have $Q_t(s, a) \geq Q^*(s, a)$, $V_t(s) \geq V^*(s)$.

Equipped with Lemma 5.1, we can decompose the regret of UCBVI-$\gamma$ as follows:

$$\text{Regret}(T) \leq \sum_{t=1}^{T} \big[V_t(s_t) - V_t^\pi(s_t)\big] = \underbrace{\sum_{t=1}^{T} \big[Q_t(s_t, a_t) - Q_t^\pi(s_t, a_t)\big]}_{\text{Regret}'(T)},$$

where the inequality holds due to Lemma 5.1. Therefore, it suffices to bound $\text{Regret}'(T)$. We have

$$\text{Regret}'(T) \leq \sum_{t=1}^{T} \Big( r(s_t, a_t) + \gamma[\mathbb{P}_{t-1} V_{t-1}](s_t, a_t) + \mathbb{C}\gamma \text{UCB}_{t-1}(s_t, a_t)$$
$$- r(s_t, a_t) - \gamma[\mathbb{P} V_{t+1}^\pi](s_t, a_t) \Big)$$
$$= \sum_{t=1}^{T} \Big( \gamma[\mathbb{P}_{t-1} V_{t-1}](s_t, a_t) + \mathbb{C}\gamma \text{UCB}_{t-1}(s_t, a_t) - \gamma[\mathbb{P} V_{t+1}^\pi](s_t, a_t) \Big),$$

where the inequality holds due to the update rule (4.2) and the Bellman equation $Q_t^\pi(s_t, a_t) = r(s_t, a_t) + \gamma[\mathbb{P} V_{t+1}^\pi](s_t, a_t)$. We further have

$$\sum_{t=1}^{T} \Big( \gamma[\mathbb{P}_{t-1} V_{t-1}](s_t, a_t) + \mathbb{C}\gamma \text{UCB}_{t-1}(s_t, a_t) - \gamma[\mathbb{P} V_{t+1}^\pi](s_t, a_t) \Big)$$

$$= \sum_{t=1}^{T} \gamma(V_{t-1}(s_{t+1}) - V_{t+1}^{\pi}(s_{t+1})) + \underbrace{\sum_{t=1}^{T} \gamma\big[(\mathbb{P}_{t-1} - \mathbb{P})(V_{t-1} - V^{*})\big](s_t, a_t)}_{I_2}$$

$$\underbrace{\phantom{\sum_{t=1}^{T} \gamma(V_{t-1}(s_{t+1}) - V_{t+1}^{\pi}(s_{t+1}))}}_{I_1}$$

$$+ \underbrace{\sum_{t=1}^{T} \gamma[(\mathbb{P}_{t-1} - \mathbb{P})V^{*}](s_t, a_t)}_{I_3} + \underbrace{\sum_{t=1}^{T} \mathbb{C}\gamma\text{UCB}_{t-1}(s_t, a_t)}_{I_4}$$

$$+ \underbrace{\sum_{t=1}^{T} \gamma\big[\mathbb{P}(V_{t-1} - V_{t+1}^{\pi})\big](s_t, a_t) - \gamma\big[V_{t-1}(s_{t+1}) - V_{t+1}^{\pi}(s_{t+1})\big]}_{I_5}. \tag{5.1}$$

In the above decomposition, term $I_1$ controls the estimation error between the value functions $V_{t-1}$ and $V_{t+1}^{\pi}$, terms $I_2$ and $I_3$ measure the estimation error between the transition probability function $\mathbb{P}$ and the estimated transition probability function $\mathbb{P}_{t-1}$, term $I_4$ comes from the exploration bonus in Algorithm 1, and term $I_5$ accounts for the randomness in the stochastic transition process, which can be controlled by the third term $O(\sqrt{TU}/(1-\gamma)^2)$ in Theorem 4.3.

In the remaining of the proof, it suffices to bound terms $I_1$ to $I_5$ separately.

First, $I_1$ can be regarded as the difference between the estimated $V_{t-1}$ and the value function $V_{t+1}^{\pi}$ of policy $\pi$, and it can be bounded by the following lemma.

**Lemma 5.2.** For the term $I_1$, We have $I_1 \leq \gamma\text{Regret}'(T) + (2S + 2)\gamma/1 - \gamma$

Next, $I_2$ can be regarded as the "correction" term between the estimated $V_{t-1}$ and the optimal value function $V^{*}$. It can be bounded by the following lemma.

**Lemma 5.3.** With probability at least $1 - 64T\delta \log^2 T/(1-\gamma)^2 - 3\delta$, we have

$$I_2 \leq (1-\gamma)\text{Regret}'(T)/2 + \sqrt{2T\mathbb{C}\log(1/\delta)} + \frac{5S^2 A\mathbb{C}\log(ST/\delta)\log(3T)}{(1-\gamma)^2}.$$

In addition, $I_3$ can be regarded as the error between the empirical probability distribution $\mathbb{P}_{t-1}$ and the true transition probability $\mathbb{P}$. Note that $V^{*}$ is a fixed value function that does not have any randomness. Therefore, $I_3$ can be bounded through the standard concentration inequalities, and its upper bound is presented in the following lemma.

**Lemma 5.4.** With probability at least $1 - 2\delta - \delta/(1-\gamma)$, we have

$$I_3 \leq \frac{2SAU^2}{1-\gamma} + U\sqrt{2SA}\sqrt{\frac{5T}{1-\gamma} + \frac{29U}{3(1-\gamma)^3} + \frac{2\text{Regret}'(T)}{1-\gamma}} + \frac{\sqrt{2TU}}{(1-\gamma)^2}.$$

Furthermore, $I_4$ can be regarded as the summation of the UCB terms, which is also the dominating term of the total regret. It can be bounded by the following lemma.

**Lemma 5.5.** With probability at least $1 - 4\delta - \delta/(1-\gamma)$, we have

$$I_4 \leq \frac{37S^2 A^{1.5}U^{3.5}}{(1-\gamma)^{2.5}} + U\sqrt{8SA}\sqrt{\frac{5T}{1-\gamma} + \frac{29U}{3(1-\gamma)^3} + \frac{2\text{Regret}'(T)}{1-\gamma}} + \frac{12SU\sqrt{AT}}{(1-\gamma)^2}.$$

Finally, $I_5$ is the summation of a martingale difference sequence. By Azuma-Hoeffding inequality, with probability at least $1 - \delta$, we have

$$I_5 \leq \frac{\sqrt{2T\log(1/\delta)}}{1-\gamma}. \tag{5.2}$$

Substituting the upper bounds of terms $I_1$ to $I_5$ from Lemma 5.2 to Lemma 5.5, as well as (5.2), into (5.1), and taking a union bound to let all the events introduced in Lemma 5.2 to Lemma 5.5 and (5.2) hold, we have with probability at least $1 - 20TU^2\delta/(1-\gamma)^2$, the following inequality holds:

$$(1-\gamma)\text{Regret}'(T) \leq \frac{160S^2 A^{1.5}U^{3.5}}{(1-\gamma)^{2.5}} + \frac{54U\sqrt{SAT}}{\sqrt{1-\gamma}} + \frac{2\sqrt{2TU}}{1-\gamma} + 12U\sqrt{\frac{SA\text{Regret}'(T)}{1-\gamma}}. \tag{5.3}$$

Using the fact that $x \leq a + b\sqrt{x} \Rightarrow x \leq 1.1a + 4b^2$, (5.3) can be further bounded as follows

$$\text{Regret}(T) \leq \text{Regret}'(T)$$
$$\leq \frac{752 S^2 A^{1.5} U^{3.5}}{(1-\gamma)^{3.5}} + \frac{60 U \sqrt{SAT}}{(1-\gamma)^{1.5}} + \frac{4\sqrt{TU}}{(1-\gamma)^2}.$$

This completes our proof.

## 5.2 Proof of Theorem 4.5

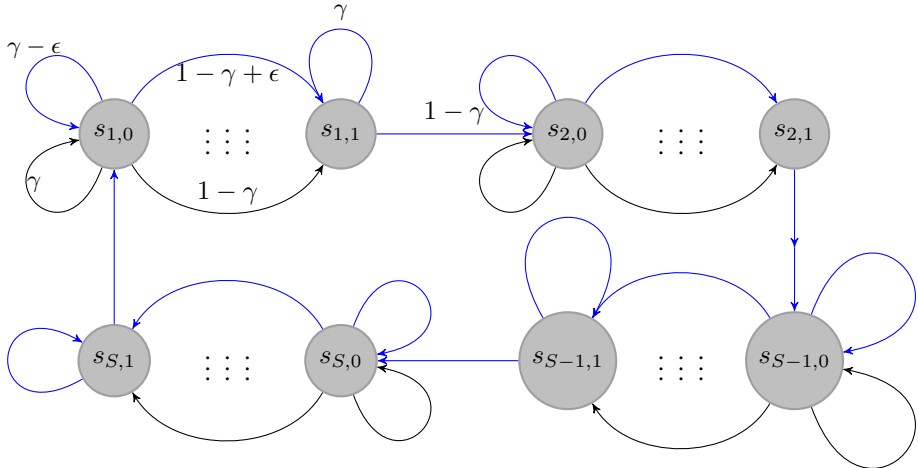

Figure 1: A class of hard-to-learn MDPs considered in Theorem 4.5. The MDP can be regarded as a combination of $S$ two-state MDPs, each of which is an MDP illustrated on the top-left corner. In addition, the $i$-th two-state MDP has the $a_i^*$-th action as its optimal action. The blue arrows represent the optimal actions in different states. $\epsilon = \sqrt{A(1-\gamma)/K}/24$.

In this subsection, we provide the proof of Theorem 4.5. The proof of the lower bound is based on constructing a class of hard MDPs. Specifically, the state space $\mathcal{S}$ consists of $2S$ states $\{s_{i,0}, s_{i,1}\}_{i \in [S]}$ and the action space $\mathcal{A}$ contains $A$ actions. The reward function $r$ satisfies that $r(s_{i,0}, a) = 0$ and $r(s_{i,1}, a) = 1$ for any $a \in \mathcal{A}, i \in [S]$. The probability transition function $\mathbb{P}$ is defined as follows.

$$\mathbb{P}(s_{i,1}|s_{i,0}, a) = 1 - \gamma + \mathbb{1}_{a=a_i^*} \frac{1}{24}\sqrt{\frac{A(1-\gamma)}{K}}, \mathbb{P}(s_{i,1}|s_{i,1}, a) = \gamma,$$

$$\mathbb{P}(s_{i,0}|s_{i,0}, a) = \gamma - \mathbb{1}_{a=a_i^*} \frac{1}{24}\sqrt{\frac{A(1-\gamma)}{K}}, \mathbb{P}(s_{i+1,0}|s_{i,1}, a) = 1 - \gamma,$$

where we assume $s_{S+1,0} = s_{1,0}$ for simplicity and $a_i^*$ is the optimal action for state $s_{i,0}$. The MDP is illustrated in Figure 1, which can be regarded as $S$ copies of the "single" two-state MDP arranged in a circle. The two-state MDP is the same as that proposed in [15]. Each of the two-state MDP has two states and one "optimal" action $a_i^*$ satisfied $\mathbb{P}(s_{i,1}|s_{i,0}, a_i^*) = 1 - \gamma + \epsilon$. Compared with the MDP instance in [10], both instances use $S$ copies of a single MDP. However, unlike the MDP in [10] which only has one "optimal" action among all $SA$ actions, our MDP which has in total $S$ "optimal" actions, which makes it harder to analyze.

Now we begin to prove our lower bound. Let $\mathbb{E}_{\mathbf{a}^*}[\cdot]$ denote the expectation conditioned on one fixed selection of $\mathbf{a}^* = (a_1^*, \ldots, a_S^*)$. We introduce a shorthand notation $\mathbb{E}^*$ to denote $\mathbb{E}^*[\cdot] = 1/A^S \cdot \sum_{\mathbf{a}^* \in \mathcal{A}^S} \mathbb{E}_{\mathbf{a}^*}[\cdot]$. Here $\mathbb{E}^*$ is the average value of expectation over the randomness from MDP defined by different optimal actions. From now on, we aim to lower bound $\mathbb{E}^*[\text{Regret}(T)]$, since once $\mathbb{E}^*[\text{Regret}(T)]$ is lower bounded, $\mathbb{E}[\text{Regret}(T)]$ can be lower bounded by selecting $a_1^*, \ldots, a_S^*$ which maximizes $\mathbb{E}[\text{Regret}(T)]$. We set $T = 10SK$ in the following proof. Based on the definition of $\mathbb{E}^*$, we have the following lemma.

**Lemma 5.6.** The expected regret $\mathbb{E}^*[\text{Regret}(T)]$ can be lower bounded as follows:

$$\mathbb{E}^*[\text{Regret}(T)] \geq \mathbb{E}^*\left[\sum_{t=1}^{T} V^*(s_t) - \frac{r(s_t, a_t)}{1-\gamma}\right] - \frac{4}{(1-\gamma)^2}.$$

By Lemma 5.6, it suffices to lower bound $\sum_{t=1}^{T}[V^*(s_t) - r(s_t, a_t)/(1-\gamma)]$, which is $\text{Regret}^{\text{Liu}}(T)$ defined in [15]. When an agent visits the state set $\{s_{j,0}, s_{j,1}\}$ for the $i$-th time, we denote the state in $\{s_{j,0}, s_{j,1}\}$ it visited as $X_{j,i}$, and the following action selected by the agent as $A_{j,i}$. Let $T_j$ be the number of steps for the agent staying in $\{s_{j,0}, s_{j,1}\}$ in the total $T$ steps. Then the regret can be further decomposed as follows:

$$\mathbb{E}^*\left[\sum_{t=1}^{T} V^*(s_t) - \frac{r(s_t, a_t)}{1-\gamma}\right] = \sum_{j=1}^{S}\mathbb{E}^*\left[\sum_{i=1}^{T_j} V^*(X_{j,i}) - \frac{r(X_{j,i}, A_{j,i})}{1-\gamma}\right] = I_1 + I_2 + I_3,$$

where

$$I_1 = \sum_{j=1}^{S}\mathbb{E}^*\left[\sum_{i=1}^{K} V^*(X_{j,i}) - \frac{r(X_{j,i}, A_{j,i})}{1-\gamma}\right],$$

$$I_2 = \sum_{j=1}^{S}\mathbb{E}^*\left[\sum_{i=K+1}^{T_j} V^*(X_{j,i}) - \frac{r(X_{j,i}, A_{j,i})}{1-\gamma}\,\middle|\, T_j > K\right] \cdot \mathbb{P}^*[T_j > K],$$

$$I_3 = -\sum_{j=1}^{S}\mathbb{E}^*\left[\sum_{i=T_j+1}^{K} V^*(X_{j,i}) - \frac{r(X_{j,i}, A_{j,i})}{1-\gamma}\,\middle|\, T_j < K\right] \cdot \mathbb{P}^*[T_j < K].$$

Note that $I_1$ essentially represents the regret over $S$ two-state MDPs in their first $K$ steps, and it can be lower bounded through the following lemma.

**Lemma 5.7.** If $K \geq 10SA/(1-\gamma)^4$, then for each $j \in [S]$, we have

$$\mathbb{E}^*\left[\sum_{i=1}^{K}(1-\gamma)V^*(X_{j,i}) - r(X_{j,i}, A_{j,i})\right] \geq \frac{\sqrt{AK}}{2304\sqrt{1-\gamma}} - \frac{1}{1-\gamma}.$$

This lemma shows that the expected regret of first $K$ steps on states $s_{j,0}$ and $s_{j,1}$ is at least $\widetilde{\Omega}\big(\sqrt{AK}/(1-\gamma)^{0.5} - 1/(1-\gamma)\big)$. Therefore by Lemma 5.7, we have

$$I_1 = \sum_{j=1}^{S}\mathbb{E}^*\left[\sum_{i=1}^{K} V^*(X_{j,i}) - \frac{r(X_{j,i}, A_{j,i})}{1-\gamma}\right] \geq \frac{\sqrt{SAT}}{2304\sqrt{10}(1-\gamma)^{1.5}} - \frac{S}{(1-\gamma)^2}. \qquad (5.4)$$

To bound $I_2$, we need the following lemma.

**Lemma 5.8.** With probability at least $1 - 2ST\delta \log T/(1-\gamma)$, for each $j \in [S]$ and $K+1 \leq t \leq T$, we have

$$\sum_{i=K+1}^{t} V^*(X_{j,i}) - \frac{r(X_{j,i}, A_{j,i})}{1-\gamma} \geq -\frac{\sqrt{2t\log(1/\delta)\log T}}{(1-\gamma)^{1.5}} - \frac{4}{(1-\gamma)^2}.$$

Lemma 5.8 gives a crude lower bound of $I_2$. Taking expectation over Lemma 5.8 and taking summation over all states, we have

$$I_2 \geq \sum_{j=1}^{S}\mathbb{E}^*\left[\left(-\frac{\sqrt{2T_j\log(1/\delta)\log T}}{(1-\gamma)^{1.5}} - \frac{4}{(1-\gamma)^2}\right)\middle|\, T_j > K\right]\mathbb{P}^*[T_j > K]$$

$$\qquad - \sum_{j=1}^{S}\frac{T}{1-\gamma}\cdot\frac{2ST\delta\log T}{(1-\gamma)^2}$$

$$\geq \sum_{j=1}^{S}\mathbb{E}^*\left[-\frac{\sqrt{2T_j\log(1/\delta)\log T}}{(1-\gamma)^{1.5}}\right] - \frac{4S}{(1-\gamma)^2} - \frac{2S^2T^2\delta\log T}{(1-\gamma)^2}$$

$$\geq \sum_{j=1}^{S}-\frac{\sqrt{2\mathbb{E}^*[T_j]\log(1/\delta)\log T}}{(1-\gamma)^{1.5}} - \frac{4S}{(1-\gamma)^2} - \frac{2S^2T^2\delta\log T}{(1-\gamma)^2}$$

$$\geq -\frac{\sqrt{2ST\log(1/\delta)\log T}}{(1-\gamma)^{1.5}} - \frac{4S}{(1-\gamma)^2} - \frac{2S^2T^2\delta\log T}{(1-\gamma)^2}, \tag{5.5}$$

where the first inequality holds due to Lemma 5.8, the second inequality holds since $1 - 2ST\delta\log T/(1-\gamma) \leq 1$ and $\mathbb{E}[-X|Y]\mathbb{P}(Y) \geq \mathbb{E}[-X]$ when $X \geq 0$, the third inequality holds due to Jensen's inequality and the fact that $\sqrt{x}$ is a concave function, and the last inequality holds due to Jensen's inequality and the fact that $\sum_{j=1}^{S} \mathbb{E}^*[T_j] = T$. To bound $I_3$, we need the following lemma, which suggests that when $K$ is large enough, $T_i > K$ happens with high probability:

**Lemma 5.9.** When $K \geq 10A\log(1/\delta)/(1-\gamma)^4$, with probability at least $1 - 2S\delta$, for all $i \in [S]$, we have $T_i > K$.

Notice that the difference of transition probability between the optimal action and suboptimal actions is $\sqrt{A(1-\gamma)}/24K$. In this case, when $T$ is large enough, $T_i$ is close to $T/S = 10K$. Thus $I_3$ can be lower bounded as follows:

$$I_3 \geq -\sum_{j=1}^{S} \frac{K}{1-\gamma}\mathbb{P}^*[T_j < K] \geq -\frac{ST\delta}{5(1-\gamma)}, \tag{5.6}$$

where the first inequality holds due to $0 \leq r(X_{j,i}, A_{j,i}) \leq 1$ and the second inequality holds due to Lemma 5.9. Finally, setting $\delta = 1/(4ST^2(1-\gamma)^2\log T)$, we can verify that the requirements of $K$ in Lemma 5.7 and Lemma 5.9 hold when $T$ satisfies $T \geq 100SAL/(1-\gamma)^4$, and $L = \log(300S^4T^2/((1-\gamma)^2\delta))\log T$. Therefore, substituting $\delta = 1/(4ST^2(1-\gamma)^2\log T)$ into (5.5) and (5.6), and combining (5.4), (5.5), (5.6) and Lemma 5.6, we have

$$\mathbb{E}[\text{Regret}(T)] \geq \frac{\sqrt{SAT}}{10000(1-\gamma)^{1.5}} - \frac{4\sqrt{STL}}{(1-\gamma)^{1.5}} - \frac{8S}{(1-\gamma)^2},$$

which completes the proof of Theorem 4.5.

# 6 Conclusions and Future Work

We proposed UCBVI-$\gamma$, an online RL algorithm for discounted tabular MDPs. We show that the regret of UCBVI-$\gamma$ can be upper bounded by $\widetilde{O}(\sqrt{SAT}/(1-\gamma)^{1.5})$ and we prove a matching lower bound on the expected regret $\widetilde{\Omega}(\sqrt{SAT}/(1-\gamma)^{1.5})$. There is still a gap between the upper and lower bounds when $T \leq \max\{S^3A^2/(1-\gamma)^4, SA/(1-\gamma)^4\}$, and we leave it as an open problem for future work.

## Acknowledgments and Disclosure of Funding

We thank Csaba Szepesvári for a valuable suggestion on improving the presentation of the proof. We thank the anonymous reviewers for their helpful comments. JH, DZ and QG are partially supported by the National Science Foundation CAREER Award 1906169, IIS-1904183, BIGDATA IIS-1855099 and AWS Machine Learning Research Award. The views and conclusions contained in this paper are those of the authors and should not be interpreted as representing any funding agencies.

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
