# A More Discussions on the Regret and Sample Complexity

## A.1 Converting Sample Complexity of Exploration to Regret

In this subsection, we shows the relationship between the sample complexity of exploration and the regret.

The definition of regret in Defintion 4.1 is related to the "sample complexity of exploration" $N(\epsilon, \delta)$ [12, 14, 9], which is the upper bound on the number of steps $t$ such that $V^*(s_t) - V_t^\pi(s_t) \geq \epsilon$ with probability at least $1 - \delta$. Compared with the regret, sample complexity of exploration focuses on the sub-optimalities at all steps $t$, rather than the first $T$ steps, and ignores the small sub-optimalities. Though both metrics have been used to describe the performance of an algorithm, these two metrics are not directly comparable. More specifically, algorithms with fewer but larger sub-optimalities will have a small sample complexity of exploration but a high regret. In contrast, algorithms with a lot of moderate sub-optimalities will have a high sample complexity of exploration but a low regret.

By the definition of the sample complexity exploration $N(\epsilon, \delta)$, with probability at least $1 - \delta$, the number of steps $t$ where $V^*(s_t) - V_t^\pi(s_t) \geq \epsilon$ is upper bounded by $N(\epsilon, \delta)$. Thus, for the regret within $T$ steps, we have following inequality:

$$
\begin{aligned}
\text{Regret}(T) &= \sum_{t=1}^{T} \left[ V^*(s_t) - V_t^\pi(s_t) \right] \\
&= \sum_{t \in [T], V^*(s_t) - V_t^\pi(s_t) \geq \epsilon} \left[ V^*(s_t) - V_t^\pi(s_t) \right] \\
&\quad + \sum_{t \in [T], V^*(s_t) - V_t^\pi(s_t) < \epsilon} \left[ V^*(s_t) - V_t^\pi(s_t) \right] \\
&\leq \frac{N(\epsilon, \delta)}{1 - \gamma} + T\epsilon,
\end{aligned}
\tag{A.1}
$$

where the inequality holds due to the definition of $N(\epsilon, \delta)$. Furthermore,if an algorithm achieve sample complexity $N(\epsilon, \delta) = O(B\epsilon^{-\alpha})$, then we can choose $\epsilon = T^{-1/(\alpha+1)}(1 - \gamma)^{1/(\alpha+1)} B^{-1/(\alpha+1)}$ to minimize the (A.1). Thus, we have

$$
\begin{aligned}
\text{Regret}(T) &\leq \frac{N(\epsilon, \delta)}{1 - \gamma} + T\epsilon \\
&= O\left( \frac{B\epsilon^{-\alpha}}{1 - \gamma} + T\epsilon \right) \\
&= O\left( B^{1/(\alpha+1)}(1 - \gamma)^{-1/(\alpha+1)} T^{\alpha/(\alpha+1)} \right).
\end{aligned}
\tag{A.2}
$$

Furthermore, the best result in sample complexity of exploration [33] achieves $\widetilde{O}\left( SA/\left((1 - \gamma)^3 \epsilon^2\right) \right)$ sample complexity and this result implies $\widetilde{O}(S^{1/3} A^{1/3} (1 - \gamma)^{-4/3} T^{2/3})$ regret, which is worse than our result by a $T^{1/6}$ factor.

## A.2 Comparison with the Regret in [15]

Our definition is similar to that of Liu and Su [15]. Note that Liu and Su [15] define the regret as $\text{Regret}^{\text{Liu}}(T) = \sum_{t=1}^{T} \Delta_t$, where $\Delta_t = (1 - \gamma)V^*(s_t) - r(s_t, a_t)$. Comparing the definition in Liu and Su [15] with our definition, we can show that $(1 - \gamma)\text{Regret}(T) \approx \text{Regret}^{\text{Liu}}(T)$ since

$$
(1 - \gamma) \sum_{t=1}^{T} V_t^\pi(s_t) \approx (1 - \gamma) \sum_{t=1}^{T} \sum_{i=0}^{\infty} \gamma^i r(s_{t+i}, a_{t+i}) \approx \sum_{t=1}^{T} r(s_t, a_t),
$$

where the first approximate equality holds due to Azuma-Hoeffding inequality and the second approximate equality holds due to $0 \leq r(s, a) \leq 1$. Therefore, our regret definition is equivalent to that in [15] up to a $1 - \gamma$ factor.

# B Proof of Lemmas in Section 5.1

In this section, we prove Lemma 5.1 to Lemma 5.5. For simplicity, we introduce the following shorthand notations:

$$\mathbb{V}^*(s,a) = \text{Var}_{s'\sim\mathbb{P}(\cdot|s,a)}\big(V^*(s')\big),$$
$$\mathbb{V}_t^\pi(s,a) = \text{Var}_{s'\sim\mathbb{P}(\cdot|s,a)}\big(V_{t+1}^\pi(s')\big),$$
$$\mathbb{V}_t(s,a) = \text{Var}_{s'\sim\mathbb{P}_t(\cdot|s,a)}(V_t(s')),$$
$$\mathbb{V}_t^*(s,a) = \text{Var}_{s'\sim\mathbb{P}_t(\cdot|s,a)}(V^*(s')).$$

We start with a list of technical lemmas that will be used to prove Lemma 5.1 to Lemma 5.5. We first provide the Azuma-Hoeffding and Bernstein inequalities.

**Lemma B.1** (Azuma–Hoeffding inequality, Cesa-Bianchi and Lugosi [6]). Let $\{x_i\}_{i=1}^n$ be a martingale difference sequence with respect to a filtration $\{\mathcal{G}_i\}$ satisfying $|x_i| \leq M$ for some constant $M$, $x_i$ is $\mathcal{G}_{i+1}$-measurable, $\mathbb{E}[x_i|\mathcal{G}_i] = 0$. Then for any $0 < \delta < 1$, with probability at least $1 - \delta$, we have

$$\sum_{i=1}^n x_i \leq M\sqrt{2n\log(1/\delta)}.$$

**Lemma B.2** (Bernstein inequality, Cesa-Bianchi and Lugosi [6]). Let $\{x_i\}_{i=1}^n$ be a martingale difference sequence with respect to a filtration $\{\mathcal{G}_i\}$ satisfying $|x_i| \leq M$ for some constant $M$, $x_i$ is $\mathcal{G}_{i+1}$-measurable, $\mathbb{E}[x_i|\mathcal{G}_i] = 0$. Suppose that

$$\sum_{i=1}^n \mathbb{E}(x_i^2|\mathcal{G}_i) \leq v$$

for some constant $v$. Then for any $\delta > 0$, with probability at least $1 - \delta$,

$$\sum_{i=1}^n x_i \leq \sqrt{2v\log(1/\delta)} + \frac{2M\log(1/\delta)}{3}.$$

The following first lemma provides basic inequalities for the summations of counted numbers $N_i(s_i, a_i)$ and $N_i(s_i)$.

**Lemma B.3.** For all $t \in [T]$ and subset $\mathcal{C} \subseteq [T]$, we have

$$\sum_{i=1}^t \frac{1}{N_{\mathbb{C}i-1}(s_i,a_i) \vee 1} \leq SA\mathbb{C}\log(3T),$$

$$\sum_{i=1}^t \frac{1}{N_{i-1}(s_i) \vee 1} \leq S\mathbb{C}\log(3T),$$

$$\sum_{i\in\mathcal{C}} \frac{1}{\sqrt{N_{i-1}(s_i,a_i) \vee 1}} \leq \sqrt{SA\mathbb{C}\log(3T)|\mathcal{C}|}.$$

Next lemma upper bounds the difference between the empirical measure $\mathbb{P}_{t-1}$ and $\mathbb{P}$, with respect to the true variance of the optimal value function $\mathbb{V}^*(s,a)$.

**Lemma B.4.** If $0 \leq V^*(s) \leq 1/(1-\gamma)$ for all $s \in \mathcal{S}$, then with probability at least $1 - \delta$, for all $t \in [T], s \in \mathcal{S}, a \in \mathcal{A}$, we have

$$\big[(\mathbb{P}_t - \mathbb{P})V^*\big](s,a) \leq \sqrt{\frac{2\mathbb{V}^*(s,a)\mathbb{C}\log(SAT/\delta)}{N_{t-1}(s,a) \vee 1}} + \frac{\mathbb{C}2\log(SAT/\delta)}{3(1-\gamma)\big(N_{t-1}(s,a) \vee 1\big)}.$$

Similar to Lemma B.4, the following lemmas also upper bounds the difference between the empirical measure $\mathbb{P}_{t-1}$ and $\mathbb{P}$, but with respect to the estimated variance.

**Lemma B.5** (Theorem 4 in Maurer and Pontil [16]). Let $Z, Z_1, .., Z_n$ be i.i.d random variable with value in $[0, M]$ and let $\delta > 0$, then with probability at least $1 - \delta$, we have

$$\mathbb{E}Z - \frac{1}{n}\sum_{i=1}^n Z_i \leq \sqrt{\frac{2\mathbb{V}_n Z\log(1/\delta)}{n}} + \frac{7M\log(1/\delta)}{3n},$$

where $\mathbb{V}_n Z$ is the estimated variance $\mathbb{V}_n Z = \sum_{1\leq i<j\leq n}(Z_i - Z_j)^2/n(n-1)$.

**Lemma B.6.** If $0 \le V^*(s) \le 1/(1-\gamma)$ for all $s \in \mathcal{S}$, then with probability at least $1 - \delta$, for all $t \in [T], s \in \mathcal{S}, a \in \mathcal{A}$, we have

$$\big[(\mathbb{P} - \mathbb{P}_t)V^*\big](s, a) \le \sqrt{\frac{2\mathbb{V}_{t-1}^*(s, a)\mathbb{C}\log(SAT/\delta)}{N_{t-1}(s, a) \vee 1}} + \frac{7\mathbb{C}\log(SAT/\delta)}{3(1 - \gamma)\big(N_{t-1}(s, a) \vee 1\big)}.$$

The next lemma shows that the total variance of the nonstationary policy $\pi$ can be upper bounded by $O(T/(1-\gamma))$. It is worth noting that a trivial bound which bounds $\mathbb{V}_i^\pi(s_i, a_i)$ by $1/(1-\gamma)^2$ only gives an $O(T/(1-\gamma)^2)$ bound.

**Lemma B.7.** With probability at least $1 - \delta/(1-\gamma)$, we have

$$\gamma^2 \sum_{t=1}^{T} \mathbb{V}_t^\pi(s_t, a_t) \le \frac{5T}{1 - \gamma} + \frac{25\log(1/\delta)}{3(1 - \gamma)^3}.$$

Based on previous concentration Lemma, we define the following high probability events and our proof of Lemma 5.2 to Lemma 5.5 relies on these high probability events. Let $\mathcal{E}$ denote the event when the conclusion of Lemma 5.1 holds. Then by Lemma 5.1, we have $\Pr(\mathcal{E}) \ge 1 - 64T\delta\log^2 T/(1-\gamma)^2$. We also define the following event:

$$\mathcal{E}_1 = \left\{ \big[(\mathbb{P}_t - \mathbb{P})V^*\big](s, a) \le \sqrt{\frac{2\mathbb{V}^*(s, a)\mathbb{C}\log(SAT/\delta)}{N_{t-1}(s, a) \vee 1}} \right.$$
$$\left. + \frac{\mathbb{C}2\log(SAT/\delta)}{3(1 - \gamma)\big(N_{t-1}(s, a) \vee 1\big)}, \forall s \in \mathcal{S}, a \in \mathcal{A}, t \in [T] \right\},$$

$$\mathcal{E}_2 = \left\{ \big[(\mathbb{P} - \mathbb{P}_t)V^*\big](s, a) \le \sqrt{\frac{2\mathbb{V}_{t-1}^*(s, a)\mathbb{C}\log(SAT/\delta)}{N_{t-1}(s, a) \vee 1}} \right.$$
$$\left. + \frac{7\mathbb{C}\log(SAT/\delta)}{3(1 - \gamma)\big(N_{t-1}(s, a) \vee 1\big)}, \forall s \in \mathcal{S}, a \in \mathcal{A}, t \in [T] \right\},$$

$$\mathcal{E}_3 = \left\{ \mathbb{P}_{t-1}(s'|s_t, a_t) - \mathbb{P}(s'|s_t, a_t) \le \sqrt{\frac{2\mathbb{P}(s'|s_t, a_t)(1 - \mathbb{P}(s'|s_t, a_t))\mathbb{C}\log(ST/\delta)}{N_{t-1}(s_t, a_t) \vee 1}}, \right.$$
$$\left. + \frac{\mathbb{C}2\log(ST/\delta)}{3\big(N_{t-1}(s_t, a_t) \vee 1\big)} \forall s \in \mathcal{S}, a \in \mathcal{A}, t \in [T] \right\},$$

$$\mathcal{E}_4 = \left\{ \sum_{t=1}^{T} \mathbb{P}(s'|s_t, a_t)\big(V_{t-1}(s') - V^*(s')\big) \le \sum_{t=1}^{T} \big(V_{t-1}(s_{t+1}) - V^*(s_{t+1})\big) + \frac{\sqrt{2T\log(1/\delta)}}{1 - \gamma} \right\},$$

$$\mathcal{E}_5 = \left\{ \gamma^2 \sum_{t=1}^{T} \mathbb{V}_t^\pi(s_t, a_t) \le \frac{5T}{1 - \gamma} + \frac{25\log(1/\delta)}{3(1 - \gamma)^3} \right\},$$

$$\mathcal{E}_6 = \left\{ \sum_{t=1}^{T} \big[\mathbb{P}(V_{t-1} - V_{t+1}^\pi)\big](s_t, a_t) - \sum_{t=1}^{T} \big[V_{t-1}(s_{t+1}) - V_{t+1}^\pi(s_{t+1})\big] \le \frac{\sqrt{2T\log(1/\delta)}}{1 - \gamma} \right\},$$

$$\mathcal{E}_7 = \left\{ \sum_{t=1}^{T} \big[\mathbb{P}(V^* - V_{t+1}^\pi)\big](s_t, a_t) - \sum_{t=1}^{T} \big[V^*(s_{t+1}) - V_{t+1}^\pi(s_{t+1})\big] \le \frac{\sqrt{2T\log(1/\delta)}}{1 - \gamma} \right\},$$

$$\mathcal{E}_8 = \left\{ \big\|\mathbb{P}_{t-1}(\cdot|s, a) - \mathbb{P}(\cdot|s, a)\big\|_1 \le \frac{\sqrt{2S\mathbb{C}\log(T/\delta)}}{\sqrt{N_{t-1}(s, a) \vee 1}}, \forall s \in \mathcal{S}, a \in \mathcal{A}, t \in [T] \right\},$$

$$\mathcal{E}_9 = \left\{ \sum_{t=1}^{T} \sum_{s'} \mathbb{P}(s'|s_t, a_t) \min \left\{ \frac{100S^2A^2U^5}{(1 - \gamma)^5\big(N_{t-1}(s') \vee 1\big)}, \frac{1}{(1 - \gamma)^2} \right\} \right.$$
$$\left. \le \sum_{t=1}^{T} \min \left\{ \frac{100S^2A^2U^5}{(1 - \gamma)^5\big(N_{t-1}(s_{t+1}) \vee 1\big)}, \frac{1}{(1 - \gamma)^2} \right\} + \frac{\sqrt{2TU}}{(1 - \gamma)^2} \right\},$$

where $U = \log(40SAT^3 \log^2 T/(\delta(1-\gamma)^2))$. For these high probability events, according to the Lemma B.1, we have $\Pr(\mathcal{E}_4) \geq 1-\delta, \Pr(\mathcal{E}_6) \geq 1-\delta, \Pr(\mathcal{E}_7) \geq 1-\delta, \Pr(\mathcal{E}_8) \geq 1-\delta, \Pr(\mathcal{E}_9) \geq 1-\delta$. According to the Lemma B.2, we have $\Pr(\mathcal{E}_3) \geq 1-\delta$. According to the Lemma B.4, we have $\Pr(\mathcal{E}_1) \geq 1-\delta$. According to the Lemma B.6, we have $\Pr(\mathcal{E}_2) \geq 1-\delta$. According to the Lemma B.7, we have $\Pr(\mathcal{E}_5) \geq 1-\delta/(1-\gamma)$.

The next lemma shows that the total difference between the optimal variance and the variance induced by $\pi$ can be bounded in terms of $\text{Regret}'(T)$.

**Lemma B.8.** On the event $\mathcal{E}_7$, we have

$$\sum_{i=1}^{T} \left( \mathbb{V}^*(s_i, a_i) - \mathbb{V}_i^\pi(s_i, a_i) \right) \leq \frac{2\text{Regret}'(T)}{1-\gamma} + \frac{2 + \sqrt{2T\mathbb{C}\log(1/\delta)}}{(1-\gamma)^2}.$$

Similar to Lemma B.8, the next lemma shows that the total difference between the estimated variance and the variance induced by $\pi$ can be upper-bounded in terms of $\text{Regret}'(T)$.

**Lemma B.9.** On the event $\mathcal{E}_6 \cap \mathcal{E}_8$, we have

$$\sum_{i=1}^{T} \left( \mathbb{V}_{i-1}(s_i, a_i) - \mathbb{V}_i^\pi(s_i, a_i) \right) \leq \frac{2\text{Regret}'(T)}{1-\gamma} + \frac{9S\sqrt{2AT\mathbb{C}\log(T/\delta)\log(3T)}}{(1-\gamma)^2}.$$

## B.1 Proof of Lemma 5.1

For simplicity, we denote $U = \log(SAT^2/\delta)$ and $H = \lfloor 2\log T/(1-\gamma) \rfloor + 1$ and for $h \in [H]$, we define

$$\text{Regret}'(t, s, h) = \sum_{1 \leq i \leq t, s_i = s} \gamma^h \left[ V_{i+h}(s_{i+h}) - V_{i+h}^\pi(s_{i+h}) \right].$$

Then we have the following lemma.

**Lemma B.10.** For each $t \in [T]$, with probability at least $1 - 4H^2\delta$, for all $s \in \mathcal{S}, h \in [H]$, we have

$$\text{Regret}'(t, s, h) \leq \frac{16SAU^2\sqrt{N_t(s)}}{(1-\gamma)^{2.5}} + \frac{4S^2A^{1.5}U^3}{(1-\gamma)^{3.5}}.$$

In addition, if $N_t(s) > 0$, we have

$$V_t(s) - V^*(s) \leq \frac{20SAU^2}{(1-\gamma)^{2.5}\sqrt{N_t(s)}}.$$

Now, we start the proof of Lemma 5.1,

*Proof of Lemma 5.1.* We prove this lemma by induction. At the first step $t = 1$, for all $s \in \mathcal{S}$, we have $V_1(s) = 1/(1-\gamma) \geq V^*(s)$. When Lemma 5.1 holds for the first $t$ steps, we consider for each $s \in \mathcal{S}, a \in \mathcal{A}$, then by the update rule (4.2), we have

$$Q_{t+1}(s, a) = \min \left\{ Q_t(s, a), r(s, a) + \gamma[\mathbb{P}_t V_t](s, a) + \mathbb{C}\gamma\text{UCB}_t(s, a) \right\}.$$

If $Q_{t+1}(s, a) = Q_t(s, a)$, then by induction, we have

$$Q_{t+1}(s, a) \geq r(s, a) + \frac{8\gamma U}{1-\gamma} \geq r(s, a) + \gamma[\mathbb{P}V^*](s, a) = Q^*(s, a),$$

where the first inequality holds due to (4.2) in Algorithm 1 and the second inequality holds due to $0 \leq V^*(s) \leq 1/(1-\gamma)$. Otherwise, if $N_t(s, a) = 0$, then we have

$$Q_{t+1}(s, a) = Q_t(s, a) \geq Q^*(s, a).$$

When $N_t(s, a) > 0$, with probability at least $1 - \delta$, we have

$Q_{t+1}(s, a) - Q^*(s, a)$
$= \gamma[\mathbb{P}_t V_t](s, a) + \mathbb{C}\gamma\text{UCB}_t(s, a) - \gamma[\mathbb{P}V^*](s, a)$

$$= \mathbb{C}\gamma \text{UCB}_t(s,a) + \gamma[(\mathbb{P}_t - \mathbb{P})V^*](s,a) + \gamma[\mathbb{P}_t(V_t - V^*)](s,a)$$

$$\geq \mathbb{C}\gamma \text{UCB}_t(s,a) + \gamma[(\mathbb{P}_t - \mathbb{P})V^*](s,a)$$

$$\geq \mathbb{C}\gamma \text{UCB}_t(s,a) - \mathbb{C}\gamma\sqrt{\frac{4\mathbb{V}_t^*(s,a)U}{N_t(s,a)\vee 1}} - \frac{8U\mathbb{C}\gamma}{(1-\gamma)(N_t(s,a)\vee 1)}$$

$$\geq \mathbb{C}\gamma\sqrt{\frac{8\mathbb{V}_t(s,a)U}{N_t(s,a)\vee 1}} - \mathbb{C}\gamma\sqrt{\frac{4\mathbb{V}_t^*(s,a)U}{N_t(s,a)\vee 1}} + \mathbb{C}\gamma\sqrt{\frac{8\sum_{s'}\mathbb{P}_t(s'|s,a)\min\{100B_t(s'),1/(1-\gamma)^2\}}{N_t(s,a)\vee 1}},$$

$$\tag{B.1}$$

where the first inequality holds due to $V_t(s) \geq V^*(s)$, the second inequality holds due to Lemma B.6 and the third inequality holds due to the definition of $\text{UCB}_t$ in (4.3). For the term $\mathbb{V}_t^*(s,a)$, we have

$$\mathbb{V}_t^*(s,a) = \mathbb{E}_{s'\sim\mathbb{P}_t(\cdot|s,a)}\left[\left(V^*(s') - \mathbb{E}[V^*(s')]\right)^2\right]$$

$$= \mathbb{E}_{s'\sim\mathbb{P}_t(\cdot|s,a)}\left[\left(V^*(s') - V_t(s') - \mathbb{E}[V^*(s') - V_t(s')] + V_t(s') - \mathbb{E}[V_t(s')]\right)^2\right]$$

$$\leq 2\mathbb{E}_{s'\sim\mathbb{P}_t(\cdot|s,a)}\left[\left(V_t(s') - \mathbb{E}[V_t(s')]\right)^2\right]$$

$$+ 2\mathbb{E}_{s'\sim\mathbb{P}_t(\cdot|s,a)}\left[\left(V^*(s') - V_t(s') - \mathbb{E}[V^*(s') - V_t(s')]\right)^2\right]$$

$$\leq 2\mathbb{V}_t(s,a) + 2\mathbb{E}_{s'\sim\mathbb{P}_t(\cdot|s,a)}\left[\left(V^*(s') - V_t(s')\right)^2\right], \tag{B.2}$$

where the first inequality holds due to $(x+y)^2 \leq 2x^2 + 2y^2$ and the second inequality holds due to $\mathbb{E}[(X - \mathbb{E}[X])^2] \leq \mathbb{E}[X^2]$. Substituting (B.2) into (B.1), with probability at least $1 - 4(t+1)H^2\delta$, we have

$$Q_{t+1}(s,a) - Q^*(s,a) \geq \mathbb{C}\gamma\sqrt{\frac{8\mathbb{V}_t(s,a)U}{N_t(s,a)\vee 1}} + \mathbb{C}\gamma\sqrt{\frac{8\sum_{s'}\mathbb{P}_t(s'|s,a)\min\{100B_t(s'),1/(1-\gamma)^2\}}{N_t(s,a)\vee 1}}$$

$$- \mathbb{C}\gamma\sqrt{\frac{8\mathbb{V}_t(s,a)U + 8U\mathbb{E}_{s'\sim\mathbb{P}_t(\cdot|s,a)}(V^*(s') - V_t(s'))^2}{N_t(s,a)\vee 1}}$$

$$\geq \mathbb{C}\gamma\sqrt{\frac{8\sum_{s'}\mathbb{P}_t(s'|s,a)\min\{100B_t(s'),1/(1-\gamma)^2\}}{N_t(s,a)\vee 1}}$$

$$- \mathbb{C}\gamma\sqrt{\frac{8U\mathbb{E}_{s'\sim\mathbb{P}_t(\cdot|s,a)}(V^*(s') - V_t(s'))^2}{N_t(s,a)\vee 1}}$$

$$\geq 0,$$

where the first inequality holds due to (B.1), the second inequality holds due to (B.2), the third inequality holds due to $\sqrt{a+b} \leq \sqrt{a} + \sqrt{b}$, the last inequality holds due to Lemma B.10 with probability at least $1 - 4H^2\delta$ and induction hypothesis with probability at least $1 - 4tH^2\delta$. In addition, for all $s \in \mathcal{S}$, we have

$$V_{t+1}(s) = \max_{a\in\mathcal{A}} Q_{t+1}(s,a) \geq \max_{a\in\mathcal{A}} Q^*(s,a) = V^*(s).$$

Thus, by induction, we complete the proof of Lemma 5.1. $\qquad\square$

### B.2 Proof of Lemma 5.2

*Proof of Lemma 5.2.* We have

$$\sum_{t=1}^{T}\gamma\left(V_{t-1}(s_{t+1}) - V_{t+1}^\pi(s_{t+1})\right)$$

$$= \gamma \sum_{t=1}^{T} \left( V_{t-1}(s_{t+1}) - V_{t+1}(s_{t+1}) \right) + \gamma \sum_{t=1}^{T} \left( V_{t+1}(s_{t+1}) - V_{t+1}^{\pi}(s_{t+1}) \right).$$

$$\underbrace{\phantom{\gamma \sum_{t=1}^{T} \left( V_{t-1}(s_{t+1}) - V_{t+1}(s_{t+1}) \right)}}_{I_1} \quad \underbrace{\phantom{\gamma \sum_{t=1}^{T} \left( V_{t+1}(s_{t+1}) - V_{t+1}^{\pi}(s_{t+1}) \right)}}_{I_2}$$

For the term $I_1$, we have

$$\sum_{t=1}^{T} \gamma \left( V_{t-1}(s_{t+1}) - V_{t+1}(s_{t+1}) \right) \leq \gamma \sum_{t=1}^{T} \sum_{s \in \mathcal{S}} \left[ V_{t-1}(s) - V_{t+1}(s) \right]$$

$$= \gamma \sum_{s \in \mathcal{S}} \sum_{t=1}^{T} \left[ V_{t-1}(s) - V_{t+1}(s) \right]$$

$$= \gamma \sum_{s \in \mathcal{S}} \left( V_0(s) + V_1(s) - V_T(s) - V_{T+1}(s) \right)$$

$$\leq \frac{2S\gamma}{1 - \gamma}, \tag{B.3}$$

where the first inequality holds due to $V_{t-1}(s) \geq V_{t+1}(s)$ by (4.2) in Algorithm 1, and the second inequality holds due to $0 \leq V_t(s) \leq 1/(1 - \gamma)$. For the term $I_2$, we have

$$I_2 = \gamma \sum_{t=2}^{T+1} \left( V_t(s_t) - V_t^{\pi}(s_t) \right)$$

$$= \gamma \text{Regret}'(T) + \gamma \left( V_{T+1}(s_{T+1}) - V_{T+1}^{\pi}(s_{T+1}) \right) - \gamma \left( V_1(s_1) - V_1^{\pi}(s_1) \right)$$

$$\leq \gamma \text{Regret}'(T) + \frac{2\gamma}{1 - \gamma}, \tag{B.4}$$

where the inequality holds due to $0 \leq V_t(s), V_t^{\pi}(s) \leq 1/(1 - \gamma)$. Combining (B.3) and (B.4), we complete the proof of Lemma 5.2. $\qquad\square$

## B.3 Proof of Lemma 5.3

*Proof of Lemma 5.3.* On the event $\mathcal{E}$, we have

$$\sum_{t=1}^{T} \gamma \left[ (\mathbb{P}_{t-1} - \mathbb{P})(V_{t-1} - V^*) \right](s_t, a_t)$$

$$= \gamma \sum_{t=1}^{T} \sum_{s' \in \mathcal{S}} \left( \mathbb{P}_{t-1}(s'|s_t, a_t) - \mathbb{P}(s'|s_t, a_t) \right) \left( V_{t-1}(s') - V^*(s') \right)$$

$$\leq \sum_{t=1}^{T} \sum_{s' \in \mathcal{S}} \left[ \sqrt{\frac{2\mathbb{P}(s'|s_t, a_t)(1 - \mathbb{P}(s'|s_t, a_t))\mathbb{C}\log(2ST/\delta)}{N_{t-1}(s_t, a_t) \vee 1}} + \frac{\mathbb{C}2\log(ST/\delta)}{3(N_{t-1}(s_t, a_t) \vee 1)} \right]$$

$$\times \left( V_{t-1}(s') - V^*(s') \right)$$

$$\leq \underbrace{\sum_{t=1}^{T} \sum_{s' \in \mathcal{S}} \sqrt{2\mathbb{C}\log(ST/\delta)} \sqrt{\frac{\mathbb{P}(s'|s_t, a_t)}{N_{t-1}(s_t, a_t) \vee 1}} \left( V_{t-1}(s') - V^*(s') \right)}_{I_1}$$

$$+ \underbrace{\sum_{t=1}^{T} \frac{2S\mathbb{C}\log(ST/\delta)}{3(1 - \gamma)(N_{t-1}(s_t, a_t) \vee 1)}}_{I_2}, \tag{B.5}$$

where first inequality holds due to the definition of $\mathcal{E}_2$ and the second inequality holds due to $0 \leq V_{t+1}(s') - V^*(s') \leq 1/(1 - \gamma)$. To bound term $I_1$, we separate $\mathcal{S}$ into two subsets $\mathcal{S}_t^1 \cup \mathcal{S}_t^2$, where

$$\mathcal{S}_t^1 = \left\{ s \in \mathcal{S} : \mathbb{P}(s|s_t, a_t)(N_{t-1}(s_t, a_t) \vee 1) \geq \frac{8\mathbb{C}\log(ST/\delta)}{(1 - \gamma)^2} \right\}, \ \mathcal{S}_t^2 = \mathcal{S}/\mathcal{S}_t^1.$$

Then on the event $\mathcal{E}_4$, we have

$$
\begin{aligned}
I_1 &= \sum_{t=1}^{T} \sum_{s' \in \mathcal{S}_t^1} \mathbb{P}(s'|s_t, a_t) \sqrt{2\mathbb{C}\log(ST/\delta)} \sqrt{\frac{1}{\mathbb{P}(s'|s_t, a_t)\big(N_{t-1}(s_t, a_t) \vee 1\big)}} \big(V_{t-1}(s') - V^*(s')\big) \\
&\quad + \sum_{t=1}^{T} \sum_{s' \in \mathcal{S}_t^2} \frac{\sqrt{2\mathbb{C}\log(ST/\delta)\mathbb{P}(s'|s_t, a_t)\big(N_{t-1}(s_t, a_t) \vee 1\big)}}{N_{t-1}(s_t, a_t) \vee 1} \big(V_{t-1}(s') - V^*(s')\big) \\
&\leq \sum_{t=1}^{T} \sum_{s' \in \mathcal{S}_t^1} (1-\gamma)\mathbb{P}(s'|s_t, a_t)\big(V_{t-1}(s') - V^*(s')\big)/2 \\
&\quad + \sum_{t=1}^{T} \sum_{s' \in \mathcal{S}_t^2} \frac{4\mathbb{C}\log(ST/\delta)}{3(1-\gamma)^2\big(N_{t-1}(s_t, a_t) \vee 1\big)} \\
&\leq \sum_{t=1}^{T} \sum_{s' \in \mathcal{S}_t^1} (1-\gamma)\mathbb{P}(s'|s_t, a_t)\big(V_{t-1}(s') - V^*(s')\big)/2 + \frac{4S^2 A\mathbb{C}\log(ST/\delta)\log(3T)}{3(1-\gamma)^2} \\
&\leq \sum_{t=1}^{T} \sum_{s' \in \mathcal{S}} (1-\gamma)\mathbb{P}(s'|s_t, a_t)\big(V_{t-1}(s') - V^*(s')\big)/2 + \frac{4S^2 A\mathbb{C}\log(ST/\delta)\log(3T)}{3(1-\gamma)^2} \\
&\leq (1-\gamma)/2 \cdot \left[\sum_{t=1}^{T} \big(V_{t-1}(s_{t+1}) - V^*(s_{t+1})\big) + \frac{\sqrt{2T\log(1/\delta)}}{1-\gamma}\right] + \frac{4S^2 A\mathbb{C}\log(ST/\delta)\log(3T)}{3(1-\gamma)^2} \\
&\leq (1-\gamma)/2 \cdot \sum_{t=1}^{T} \big(V_{t-1}(s_{t+1}) - V_{t+1}^{\pi}(s_{t+1})\big) + \sqrt{2T\mathbb{C}\log(1/\delta)} + \frac{4S^2 A\mathbb{C}\log(ST/\delta)\log(3T)}{3(1-\gamma)^2} \\
&\leq (1-\gamma)/2 \cdot \left[\text{Regret}'(T) + \frac{(2S+2)}{1-\gamma}\right] + \sqrt{2T\mathbb{C}\log(1/\delta)} + \frac{4S^2 A\mathbb{C}\log(ST/\delta)\log(3T)}{3(1-\gamma)^2},
\end{aligned}
$$
(B.6)

where the first inequality holds due to separate condition of $\mathbb{P}(s')$, the second inequality holds due to Lemma B.3, the third inequality holds due to $V_{t-1}(s') \geq V^*(s')$, the fourth inequality holds due to the definition of event $\mathcal{E}_4$, the fifth inequality holds due to $V^* \geq V_{t+1}^{\pi}$, and the last inequality holds due to Lemma 5.2. For the term $I_2$, according to Lemma B.3, we have

$$
I_2 \leq \frac{2S^2 A\mathbb{C}\log(ST/\delta)\log(3T)}{3(1-\gamma)}.
$$
(B.7)

Substituting (B.6),(B.7) into (B.5), we complete the proof of Lemma 5.3. □

## B.4  Proof of Lemma 5.4

*Proof of Lemma 5.4.* On the event $\mathcal{E}_1 \cap \mathcal{E}_5 \cap \mathcal{E}_7$, we have

$$
\begin{aligned}
\sum_{t=1}^{T} &\gamma[(\mathbb{P}_{t-1} - \mathbb{P})V^*](s_t, a_t) \\
&\leq \sum_{t=1}^{T} \mathbb{C}\gamma \sqrt{\frac{2\mathbb{V}^*(s_t, a_t)\mathbb{C}\log(SAT/\delta)}{N_{t-1}(s_t, a_t) \vee 1}} + \frac{2\mathbb{C}\log(SAT/\delta)\mathbb{C}\gamma}{(1-\gamma)\big(N_{t-1}(s_t, a_t) \vee 1\big)} \\
&\leq \mathbb{C}\gamma\sqrt{2\mathbb{C}\log(SAT/\delta)} \sqrt{\sum_{t=1}^{T} \mathbb{V}^*(s_t, a_t)} \sqrt{\sum_{t=1}^{T} \frac{1}{N_{t-1}(s_t, a_t) \vee 1}} \\
&\quad + \sum_{t=1}^{T} \frac{2\gamma\mathbb{C}\log(SAT/\delta)}{(1-\gamma)\big(N_{t-1}(s_t, a_t) \vee 1\big)}
\end{aligned}
$$

$$\leq \mathbb{C}\gamma U\sqrt{2SA}\sqrt{\sum_{t=1}^{T}\mathbb{V}^*(s_t,a_t)+\frac{2\gamma SAU^2}{1-\gamma}}$$

$$=\mathbb{C}\gamma U\sqrt{2SA}\sqrt{\sum_{t=1}^{T}\mathbb{V}_t^{\pi}(s_t,a_t)+\sum_{t=1}^{T}\mathbb{V}^*(s_t,a_t)-\sum_{t=1}^{T}\mathbb{V}_t^{\pi}(s_t,a_t)+\frac{2\gamma SAU^2}{1-\gamma}}$$

$$\leq U\sqrt{2SA}\sqrt{\frac{5T}{1-\gamma}+\frac{29U}{3(1-\gamma)^3}+\frac{2\text{Regret}'(T)}{1-\gamma}+\frac{\sqrt{2TU}}{(1-\gamma)^2}+\frac{2SAU^2}{1-\gamma}}, \tag{B.8}$$

where the first inequality holds due to the definition of event $\mathcal{E}_1$, the second inequality holds due to Cauchy-Schwarz inequality, the third inequality holds due to Lemma B.3 and the definition of $U$, and the last inequality holds due to Lemma B.8 and the definition of event $\mathcal{E}_5$. Thus, we complete the proof of Lemma 5.4. □

## B.5 Proof of Lemma 5.5

*Proof of Lemma 5.5.* For the term $\text{UCB}_{t-1}(s_t,a_t)$, we have

$$\sum_{t=1}^{T}\mathbb{C}\gamma\text{UCB}_{t-1}(s_t,a_t)\leq\underbrace{\sum_{t=1}^{T}\mathbb{C}\gamma\sqrt{\frac{8U\mathbb{V}_{t-1}(s_t,a_t)}{N_{t-1}(s_t,a_t)\vee 1}}}_{I_1}+\underbrace{\sum_{t=1}^{T}\mathbb{C}\gamma\frac{8U}{(1-\gamma)\big(N_{t-1}(s_t,a_t)\vee 1\big)}}_{I_2}$$

$$+\underbrace{\sum_{t=1}^{T}\mathbb{C}\gamma\sqrt{\frac{8\sum_{s'}\mathbb{P}_t(s'|s_t,a_t)\min\big\{100B_t(s'),1/(1-\gamma)^2\big\}}{N_{t-1}(s_t,a_t)\vee 1}}}_{I_3}.$$

$$\tag{B.9}$$

For the term $I_1$, on the event $\mathcal{E}_5\cap\mathcal{E}_6\cap\mathcal{E}_8$, we have

$$I_1\leq\mathbb{C}\gamma\sqrt{8U\sum_{t=1}^{T}\mathbb{V}_{t-1}(s_t,a_t)}\sqrt{\sum_{t=1}^{T}\frac{1}{N_{t-1}(s_t,a_t)\vee 1}}$$

$$\leq\mathbb{C}\gamma U\sqrt{8SA}\sqrt{\sum_{t=1}^{T}\mathbb{V}_{t-1}(s_t,a_t)}$$

$$=\mathbb{C}\gamma U\sqrt{8SA}\sqrt{\sum_{i=1}^{T}\mathbb{V}_t^{\pi}(s_t,a_t)+\sum_{t=1}^{T}\mathbb{V}_{t-1}(s_t,a_t)-\sum_{i=1}^{T}\mathbb{V}_t^{\pi}(s_t,a_t)}$$

$$\leq U\sqrt{8SA}\sqrt{\frac{5T}{1-\gamma}+\frac{29U}{3(1-\gamma)^3}+\frac{2\text{Regret}'(T)}{1-\gamma}+\frac{9SU\sqrt{AT}}{(1-\gamma)^2}}, \tag{B.10}$$

where the first inequality holds due to Cauchy-Schwarz inequality, the second inequality holds due to Lemma B.3, the last inequality holds due to the definition of event $\mathcal{E}_5$ and Lemma B.9. For the term $I_2$, by Lemma B.3, we have

$$I_2=\sum_{t=1}^{T}\frac{8U}{(1-\gamma)\big(N_{t-1}(s_t,a_t)\vee 1\big)}\leq\frac{8SAU^2}{1-\gamma}. \tag{B.11}$$

For the term $I_3$, on the event $\mathcal{E}_8\cap\mathcal{E}_9$, we have

$I_3$

$$\leq\sqrt{8\sum_{t=1}^{T}\frac{1}{N_{t-1}(s_t,a_t)\vee 1}}\sqrt{\sum_{t=1}^{T}\sum_{s'}\mathbb{P}_t(s'|s_t,a_t)\min\left\{\frac{100S^2A^2U^5}{(1-\gamma)^5N_{t-1}(s')},\frac{1}{(1-\gamma)^2}\right\}}$$

$$\leq \sqrt{8SAU}\sqrt{\sum_{t=1}^{T}\sum_{s'}\mathbb{P}_t(s'|s_t, a_t)\min\left\{\frac{100S^2A^2U^5}{(1-\gamma)^5(N_{t-1}(s')\vee 1)}, \frac{1}{(1-\gamma)^2}\right\}}$$

$$\leq \sqrt{8SAU}\cdot$$

$$\sqrt{\sum_{i=1}^{T}\frac{\sqrt{2SU}}{(1-\gamma)^2\sqrt{N_t(s_t,a_t)\vee 1}} + \sum_{t=1}^{T}\sum_{s'}\mathbb{P}(s'|s_t,a_t)\min\left\{\frac{100S^2A^2U^5}{(1-\gamma)^5(N_{t-1}(s')\vee 1)}, \frac{1}{(1-\gamma)^2}\right\}}$$

$$\leq \sqrt{8SAU}\sqrt{\frac{SU\sqrt{2AT}}{(1-\gamma)^2} + \frac{\sqrt{2TU}}{(1-\gamma)^2} + \sum_{t=1}^{T}\min\left\{\frac{100S^2A^2U^5}{(1-\gamma)^5(N_{t-1}(s_{t+1})\vee 1)}, \frac{1}{(1-\gamma)^2}\right\}}$$

$$\leq \sqrt{8SAU}\sqrt{\frac{SU\sqrt{2AT}}{(1-\gamma)^2} + \frac{\sqrt{2TU}}{(1-\gamma)^2} + \frac{100S^3A^2U^6}{(1-\gamma)^5}}, \tag{B.12}$$

where the first inequality holds due to Cauchy-Schwarz inequality, the second inequality holds due to Lemma B.3, the third inequality holds due to the definition of event $\mathcal{E}_8$, the forth inequality holds due to the definition of event $\mathcal{E}_9$ and the last inequality holds due to Lemma B.3. Substituting (B.10), (B.11) and (B.12) into (B.9), we complete the proof of Lemma 5.5. □

## C  Proof of Lemmas in Section 5.2

### C.1  Proof of Lemma 5.6

*Proof of Lemma 5.6.* We have

$$\mathbb{E}^*\left[\sum_{t=1}^{T}V^*(s_t) - V_t^\pi(s_t)\right] = \mathbb{E}^*\left[\sum_{t=1}^{T}V^*(s_t) - \sum_{k=0}^{\infty}\gamma^k r(s_{t+k}, a_{t+k})\right]$$

$$= \mathbb{E}^*\left[\sum_{t=1}^{T}\left(V^*(s_t) - \sum_{k=0}^{t}\gamma^k r(s_t, a_t)\right) - \sum_{t=T+1}^{\infty}\sum_{k=0}^{T}\gamma^{t-k}r(s_t, a_t)\right]$$

$$\geq \mathbb{E}^*\left[\sum_{t=1}^{T}V^*(s_t) - \frac{r(s_t, a_t)}{1-\gamma}\right] - \sum_{t=T+1}^{\infty}\sum_{k=0}^{T}\gamma^{t-k}$$

$$\geq \mathbb{E}^*\left[\sum_{t=1}^{T}V^*(s_t) - \frac{r(s_t, a_t)}{1-\gamma}\right] - \frac{4}{(1-\gamma)^2}. \tag{C.1}$$

where the first inequality holds due to $0 \leq r(s_t, a_t) \leq 1$ and the last inequality holds due to $\sum_{k=0}^{\infty}\gamma^k = 1/(1-\gamma)$. Thus, we finish the proof of Lemma 5.6. □

### C.2  Proof of Lemma 5.7

*Proof of Lemma 5.7.* In this proof, we follow the proof technique in [15] and [10]. For simplicity, we denote $\epsilon = \sqrt{A(1-\gamma)/K}/24$ and we first determine the optimal policy in these hard-to-learn MDPs. According to (3.1), for optimal policy $\pi^*$, we have

$$Q^*(s, a) = r(s, a) + \gamma[\mathbb{P}V^*](s, a),$$

For each $j \in [S]$ and state $s = s_{j,1}$, the choice of action $a$ will not effect the reward $r(s, a)$ and the probability transition function $\mathbb{P}(\cdot|s, a)$. For optimal action $a^*$ at state $s = s_{j,0}$, we have

$$V^*(s_{j,0}) = r(s, a) + \gamma[\mathbb{P}V^*](s, a^*)$$
$$= 0 + \gamma\mathbb{P}(s_{j,0}|s_{j,0}, a^*)V^*(s_{j,0}) + \gamma\mathbb{P}(s_{j,1}|s_{j,0}, a^*)V^*(s_{j,1}).$$

Since $\mathbb{P}(s_{j,0}|s_{j,0}, a^*) + \mathbb{P}(s_{j,1}|s_{j,0}, a^*) = 1$, we have

$$(1-\gamma)V^*(s_{j,0}) = \gamma\big(V^*(s_{j,1}) - V^*(s_{j,0})\big),$$

and it implies that $V^*(s_{j,1}) \geq V^*(s_{j,0})$. Therefore, for all action $a \neq a_j^*$, we have $Q^*(s_{j,0}, a_j^*) \geq Q^*(s_{j,0}, a)$ and it further implies that the optimal action at state $s = s_{j,0}$ is $a_j^*$. Thus, according to the optimal bellman equation 3.1, for each $j \in [S]$, we have

$$V^*(s_{j,0}) = \gamma(1 - \gamma + \epsilon)V^*(s_{j,1}) + \gamma(\gamma - \epsilon)V^*(s_{j,0}),$$
$$V^*(s_{j,1}) = 1 + \gamma(1 - \gamma)V^*(s_{j+1,1}) + \gamma^2 V^*(s_{j,1}),$$

and it implies that the optimal value function $V^*$ is

$$V^*(s_{j,0}) = \frac{\gamma - \gamma^2 + \gamma\epsilon}{(1 - \gamma)(1 - 2\gamma^2 + \gamma + \gamma\epsilon)},$$
$$V^*(s_{j,1}) = \frac{1 - \gamma^2 + \gamma\epsilon}{(1 - \gamma)(1 - 2\gamma^2 + \gamma + \gamma\epsilon)}.$$

When an agent visits the state set $\{s_{j,0}, s_{j,1}\}$ for the $i$-th time, we denote the state in $\{s_{j,0}, s_{j,1}\}$ it visited as $X_{j,i}$, and the following action selected by the agent as $A_{j,i}$. For each $j \in [S]$, by the definition of $X_{j,i}$, we have

$$\mathbb{P}(X_{j,i} = s_{j,1}|X_{j,i-1} = s_{j,0}, A_{j,i-1}) = 1 - \gamma + \mathbb{1}_{A_{j,i}=a_j^*}\,\epsilon,$$
$$\mathbb{P}(X_{j,i} = s_{j,0}|X_{j,i-1} = s_{j,0}, A_{j,i-1}) = \gamma - \mathbb{1}_{a=a_j^*}\,\epsilon,$$
$$\mathbb{P}(X_{j,i} = s_{j,0}|X_{j,i-1} = s_{j,0}, A_{j,i-1}) = 1 - \gamma,$$
$$\mathbb{P}(X_{j,i} = s_{j,1}|X_{j,i-1} = s_{j,1}, A_{j,i-1}) = \gamma,$$

where the third equality holds because when $X_{j,i-1}$ leave state $s_{j,0}, s_{j,1}$, the next state in $s_{j,0}, s_{j,1}$ must be $s_{j,0}$. Similar to the proof of Theorem 5 in [10], we focus on the first $K$ visits to the state set $\{s_{j,0}, s_{j,1}\}$ and let random variable $N_0, N_1$ and $N_0^*$ denote the total number of visit state $s_{j,0}$, the total number of visit state $s_{j,1}$ and the total number of visit state $s_{j,0}$ with action $a_j^*$. By the same argument as the proof of Theorem 5 in [10], for the random variable $N_1$ and $N_0^*$, we have following property:

$$\mathbb{E}[N_1] \leq \frac{K}{2} + \frac{1}{2(1 - \gamma)} + \frac{\epsilon\mathbb{E}[N_0^*]}{1 - \gamma}, \tag{C.2}$$

and

$$\mathbb{E}[N_0^*] \leq \frac{K}{2A} + \frac{1}{2A(1 - \gamma)} + \frac{\epsilon K}{2}\sqrt{\frac{K}{A(1 - \gamma)}} + \frac{\epsilon K}{2\sqrt{A}(1 - \gamma)}. \tag{C.3}$$

Therefore, the regret can be upper bounded by

$$\mathbb{E}^*\left[\sum_{i=1}^K V^*(X_{j,i}) - \frac{r(X_{j,i}, A_{j,i})}{1 - \gamma}\right]$$

$$= \mathbb{E}[N_0]\big(V^*(s_{j,0}) - 0\big) + \mathbb{E}[N_1]\left(V^*(s_{j,1}) - \frac{1}{1 - \gamma}\right)$$

$$= \frac{(\gamma - \gamma^2 + \gamma\epsilon)\big(K - \mathbb{E}[N_1]\big) - (\gamma - \gamma^2)\mathbb{E}[N_1]}{(1 - \gamma)(1 - 2\gamma^2 + \gamma + \gamma\epsilon)}$$

$$\geq \frac{\frac{K\gamma\epsilon}{2} - \gamma - \frac{\gamma\epsilon}{2(1-\gamma)} - \frac{\mathbb{E}[N_0^*]\epsilon(2\gamma - 2\gamma^2 + \gamma\epsilon)}{1-\gamma}}{(1 - \gamma)(1 - 2\gamma^2 + \gamma + \gamma\epsilon)}$$

$$\geq \frac{\frac{K\gamma\epsilon}{2} - \gamma - \frac{\gamma\epsilon}{2(1-\gamma)} - \left(\frac{K}{2A} + \frac{1}{2A(1-\gamma)} + \frac{\epsilon K}{2}\sqrt{\frac{K}{A(1-\gamma)}} + \frac{\epsilon K}{2\sqrt{A}(1-\gamma)}\right)\frac{\epsilon(2\gamma - 2\gamma^2 + \gamma\epsilon)}{1-\gamma}}{(1 - \gamma)(1 - 2\gamma^2 + \gamma + \gamma\epsilon)}. \tag{C.4}$$

where the second inequality holds due to the fact that $\mathbb{E}[N_0] + \mathbb{E}[N_1] = K$, the third inequality holds due to (C.2) and the last inequality holds due to (C.3). Since $K \geq 10SA/(1 - \gamma)^4$, $\gamma > 2/3$ and $A \geq 30$, (C.4) can be further bounded by

$$\mathbb{E}^*\left[\sum_{i=1}^K V^*(X_{j,i}) - \frac{r(X_{j,i}, A_{j,i})}{1 - \gamma}\right]$$

$$\geq \frac{\frac{K\gamma\epsilon}{2} - \gamma - \frac{\gamma\epsilon}{2(1-\gamma)} - \left( \frac{K}{2A} + \frac{1}{2A(1-\gamma)} + \frac{\epsilon K}{2}\sqrt{\frac{K}{A(1-\gamma)}} + \frac{\epsilon K}{2\sqrt{A(1-\gamma)}} \right) \frac{\epsilon(2\gamma - 2\gamma^2 + \gamma\epsilon)}{1-\gamma}}{(1-\gamma)(1 - 2\gamma^2 + \gamma + \gamma\epsilon)}$$

$$\geq \gamma \times \frac{\frac{K\epsilon}{4} - 1 - 3\epsilon \left( \frac{5K}{8A} + \frac{\epsilon K}{2}\sqrt{\frac{K}{A(1-\gamma)}} + \frac{\epsilon K}{2\sqrt{A(1-\gamma)}} \right)}{(1-\gamma)(1 - 2\gamma^2 + \gamma + \gamma\epsilon)}$$

$$\geq \gamma \times \frac{\frac{\sqrt{AK(1-\gamma)}}{576} - 1}{(1-\gamma)(1 - 2\gamma^2 + \gamma + \gamma\epsilon)}$$

$$\geq \frac{\sqrt{AK}}{2304(1-\gamma)^{1.5}} - \frac{1}{(1-\gamma)^2}, \tag{C.5}$$

where the second inequality holds to $\epsilon = \sqrt{A(1-\gamma)/K}/24 \leq 1 - \gamma$ with $K \geq 10SA/(1-\gamma)^4$, the third inequality holds due to $\epsilon = \sqrt{A(1-\gamma)/K}/24$ with $A \geq 30$ and the last inequality holds due to $\gamma \geq 2/3$ and $\epsilon = \sqrt{A(1-\gamma)/K}/24 \leq 1 - \gamma$. Therefore, we finish the proof of Lemma 5.7.

$\square$

## C.3 Proof of Lemma 5.8

*Proof of Lemma 5.8.* For each $j \in [S]$ and $t \in [T]$, we denote $H = \lfloor \log T/(1-\gamma) \rfloor + 1$, random variable

$$Y_{j,i} = \sum_{k=0}^{H} \gamma^k r(X_{j,i+k}, A_{j,i+k}),$$

and filtration $\mathcal{F}_{j,i}$ contain all random variable before $X_{j,i+H}$. For simplicity, we ignore the subscript $j$ and only focus on the subscript $i$.

Since $Y_i$ is $\mathcal{F}_i$-measurable and $0 \leq Y_i \leq 1/(1-\gamma)$, for each $k \in [H]$, with probability at least $1 - \delta$, we have

$$\sum_{i=\lfloor \frac{K}{H} \rfloor+1}^{\lfloor \frac{t}{H} \rfloor+1} Y_{iH+k} \leq \sum_{i=\lfloor \frac{K}{H} \rfloor+1}^{\lfloor \frac{t}{H} \rfloor+1} \mathbb{E}\left[ Y_{iH+k} | \mathcal{F}_{(i-1)H+k} \right] + \sqrt{\frac{2t}{1-\gamma} \log \frac{1}{\delta}}$$

$$= \sum_{i=\lfloor \frac{K}{H} \rfloor+1}^{\lfloor \frac{t}{H} \rfloor+1} V_{iH+k}^{\pi}(X_{iH+k}) + \sqrt{\frac{2t}{1-\gamma} \log \frac{1}{\delta}}$$

$$\leq \sum_{i=\lfloor \frac{K}{H} \rfloor+1}^{\lfloor \frac{t}{H} \rfloor+1} V^*(X_{iH+k}) + \sqrt{\frac{2t}{1-\gamma} \log \frac{1}{\delta}}, \tag{C.6}$$

where the first inequality holds due to Lemma B.1 and the second inequality holds due to the definition of optimal value function $V^*$. Taking summation of (C.6), for all $k \in [H]$, with probability at least $1 - H\delta$, we have

$$\sum_{i=K+1}^{t} V^*(X_i) + \frac{\sqrt{2t \log \frac{1}{\delta} \log T}}{(1-\gamma)^{1.5}} \geq \sum_{i=K+1}^{t} Y_i$$

$$= \sum_{i=K+1}^{t} \sum_{k=0}^{H} \gamma^k r(X_{i+k}, A_{i+k})$$

$$\geq \sum_{i=K+1}^{t} r(X_i, A_i) \sum_{k=0}^{\min(H, i-K-1)} \gamma^i$$

$$\geq \sum_{i=K+1}^{t} \frac{r(X_i, A_i)}{1-\gamma} - \frac{4}{(1-\gamma)^2},$$

where the second inequality holds due to $0 \leq r(s,a) \leq 1$. Finally, taking union for all $j \in [S]$ and $t \in [T]$, we complete the proof. $\square$

## C.4 Proof of Lemma 5.9

*Proof of Lemma 5.9.* Let $Y_{j,i}$ be an indicator random variables which denote whether the agent at state $X_{j,i}$ with action $A_{j,i}$ goes to the different state. $Y_{j,i} = 1$ if the agent goes to the different state and $Y_{j,i} = 0$ if the agent stay at the same state. Let filtration $\mathcal{F}_{j,i}$ contain all random variables before $X_{j,i}$. Then, for each $j \in [S]$, with probability at least $1 - \delta$, we have

$$\sum_{i=1}^{K} Y_{j,i} \leq \sum_{i=1}^{K} \mathbb{E}\big[Y_{j,i}|\mathcal{F}_{j,i-1}\big] + \sqrt{2K \log \frac{1}{\delta}} \leq (1 - \gamma + \epsilon)K + \sqrt{2K \log \frac{1}{\delta}} \leq 3(1 - \gamma)K, \tag{C.7}$$

where the first inequality holds due to Lemma B.1, the second inequality holds due to the definition of our MDPs and the last one holds due to the selection of $K$. Similarly, with probability at least $1 - \delta$, we have

$$\sum_{i=1}^{5K} Y_{j,i} \geq \sum_{i=1}^{2K} \mathbb{E}\big[Y_{j,i}|\mathcal{F}_{j,i-1}\big] - \sqrt{10K \log \frac{1}{\delta}} \geq 5K(1 - \gamma) - \sqrt{10K \log \frac{1}{\delta}} \geq 4(1 - \gamma)K, \tag{C.8}$$

where the first inequality holds due to Lemma B.1, the second inequality holds due to the definition of our MDPs and the last one holds due to the selection of $K$. Taking a union bound (C.7) and (C.8) for all $j \in [S]$, then we have (C.7) and (C.8) hold with probability at least $1 - 2S\delta$. Let $Z_{j,i}$ be the number of times for the agent to start from state $s_{j,i}$ and travel the next different state in the first $T$ steps. By definition, we have

$$Z_{j,0} + Z_{j,1} = \sum_{i=1}^{T_j} Y_{j,i}. \tag{C.9}$$

By Pigeonhole principle, there exist a $j^*$ such that $T_{j^*} \geq T/S = 10K > 5K$. Therefore, we have

$$Z_{j^*,0} + Z_{j^*,1} = \sum_{i=1}^{T_{j^*}} Y_{j^*,i} \geq \sum_{i=1}^{5K} Y_{j^*,i} \geq 4(1 - \gamma)K. \tag{C.10}$$

Furthermore, after leaving the state $s_{j^*,0}$, the agent will visit all other states before arrive the state $s_{j^*,0}$ again. Thus, for any $k \in [S]$, the difference between $Z_{j^*,0}$ and $Z_{k,0}$ is at most 1, so do $Z_{j^*,1}$ and $Z_{k,1}$. Therefore, for any $k \in [S]$, we have

$$Z_{k,0} + Z_{k,1} \geq Z_{j^*,0} + Z_{j^*,1} - 2 \geq 4(1 - \gamma)K - 2 > 3(1 - \gamma)K \geq \sum_{i=1}^{K} Y_{k,i}, \tag{C.11}$$

where the second inequality holds due to (C.10), the third inequality holds since $K > 2/(1 - \gamma)$ and the last one holds due to (C.7). Finally, by (C.9) we have $Z_{k,0} + Z_{k,1} = \sum_{i=1}^{T_k} Y_{k,i}$. Combining it with (C.11), we have $\sum_{i=1}^{T_k} Y_{k,i} > \sum_{i=1}^{K} Y_{k,i}$, which suggests that $T_k > k$. Thus, we complete the proof.

$\square$

# D Proof of Lemmas in Appendix B

## D.1 Proof of Lemma B.3

*Proof of Lemma B.3.* We have

$$\sum_{i=1}^{t} \frac{1}{N_{i-1}(s_i, a_i) \vee 1} = \sum_{s \in \mathcal{S}, a \in \mathcal{A}} 1 + \sum_{s \in \mathcal{S}, a \in \mathcal{A}} \sum_{i=1}^{N_{t-1}(s,a)} \frac{1}{i} \leq SA + \sum_{s \in \mathcal{S}, a \in \mathcal{A}} \sum_{i=1}^{t} \frac{1}{i} \leq SA\mathbb{C}\log(3T). \tag{D.1}$$

We also have

$$\sum_{i=1}^{t} \frac{1}{N_{i-1}(s_i) \vee 1} = \sum_{s \in \mathcal{S}} 1 + \sum_{i=1}^{N_t(s)} \frac{1}{i} \leq S + \sum_{s \in \mathcal{S}} \sum_{i=1}^{t} \frac{1}{i} \leq S\mathbb{C}\log(3T).$$

According to (D.1), for a subset $\mathcal{C} \subseteq [T]$, we have

$$\sum_{i \in \mathcal{C}} \frac{1}{\sqrt{N_{i-1}(s_i, a_i) \vee 1}} \leq \sqrt{|\mathcal{C}| \sum_{i \in \mathcal{C}} \frac{1}{N_{i-1}(s_i, a_i) \vee 1}} \leq \sqrt{SA\mathbb{C}\log(3T)|\mathcal{C}|},$$

where the first inequality holds due to Cauchy-Schwarz inequality and the second inequality holds due to (D.1). Thus, we complete the proof. $\qquad\square$

## D.2  Proof of Lemma B.4

*Proof of Lemma B.4.* For each $s \in \mathcal{S}, a \in \mathcal{A}$, we denote $t_0 = 0$ and

$$t_i = \min\{t | t > t_{i-1}, (s_t, a_t) = (s, a)\}. \tag{D.2}$$

Here, $t_i$ is the time which state-action pair $(s, a)$ appear for the $i$th time and the random variable $t_i$ is a stopping time. Beside, the random variable $V^*(s_{t_i+1})(i = 1, 2., ,)$ are random variable with value in $[0, 1/(1 - \gamma)]$ and variance $\mathbb{V}^*(s, a)$. By Lemma B.2 and a union bound, with probability at least $1 - \delta$, for all $s \in \mathcal{S}, a \in \mathcal{A}, \tau \in [T]$, we have

$$\sum_{i=1}^{\tau} V^*(s_{t_i+1}) - \sum_{i=1}^{\tau} \mathbb{P}V^*(s, a) \leq \sqrt{2\tau \mathbb{V}^*(s, a) \log(SAT/\delta)} + \frac{2\log(SAT/\delta)}{3(1 - \gamma)}.$$

Thus, for all $\tau \in [T]$, we have

$$\begin{aligned}
\left[(\mathbb{P}_{t_\tau+1} - \mathbb{P})V^*\right](s, a) &= \frac{1}{\tau} \sum_{i=1}^{\tau} V^*(s_{t_i+1}) - \frac{1}{\tau} \sum_{i=1}^{\tau} \mathbb{P}V^*(s, a) \\
&\leq \sqrt{\frac{2\mathbb{V}^*(s, a)\mathbb{C}\log(SAT/\delta)}{\tau}} + \frac{\mathbb{C}2\log(SAT/\delta)}{3(1 - \gamma)\tau} \\
&= \sqrt{\frac{2\mathbb{V}^*(s, a)\mathbb{C}\log(SAT/\delta)}{N_{t_\tau}(s, a)}} + \frac{\mathbb{C}2\log(SAT/\delta)}{3(1 - \gamma)N_{t_\tau}(s, a)}. \tag{D.3}
\end{aligned}$$

In addition, for $\tau = 0$, we have

$$\left[(\mathbb{P}_{t_\tau+1} - \mathbb{P})V^*\right](s, a) \leq \frac{1}{1 - \gamma} \leq \frac{\mathbb{C}2\log(SAT/\delta)}{3(1 - \gamma)(N_{t_\tau}(s, a) \vee 1)}, \tag{D.4}$$

where the first inequality holds due to $0 \leq V^*(s) \leq 1/(1 - \gamma)$ and the second inequality holds due to $N_{t_\tau}(s, a) = 0$. Since $\mathbb{P}_t$ and $N_{t-1}(s, a)$ changed only when $t = t_\tau + 1$, we complete the proof by combining (D.3) and (D.4). $\qquad\square$

## D.3  Proof of Lemma B.6

*Proof of Lemma B.6.* For each $s \in \mathcal{S}, a \in \mathcal{A}$, we denote $t_0 = 0$ and denote

$$t_i = \min\{t | t > t_{i-1}, (s_t, a_t) = (s, a)\}. \tag{D.5}$$

Here, $t_i$ is the time which state-action pair $(s, a)$ appear for the $i$th time and the random variable $t_i$ is a stopping time. Beside, the random variable $V^*(s_{t_i+1})(i = 1, 2., ,)$ are random variable with value in $[0, 1/(1 - \gamma)]$ and variance $\mathbb{V}^*(s, a)$. By Lemma B.5 and a union bound, with probability at least $1 - \delta$, for all $s \in \mathcal{S}, a \in \mathcal{A}, \tau \in [T]$, we have

$$\sum_{i=1}^{\tau} \mathbb{P}V_t^*(s, a) - \sum_{i=1}^{\tau} V^*(s_{t_i+1}) \leq \sqrt{2\tau \mathbb{V}_{t_\tau}^*(s, a) \log(SAT/\delta)} + \frac{7\log(SAT/\delta)}{3(1 - \gamma)}.$$

Thus, for all $\tau \in [T]$, we have

$$
\begin{aligned}
\left[(\mathbb{P} - \mathbb{P}_{t_\tau+1})V^*\right](s,a) &= \frac{1}{\tau}\Big|\sum_{i=1}^\tau V^*(s_{t_i+1}) - \sum_{i=1}^\tau \mathbb{P}V^*(s,a)\Big| \\
&\leq \sqrt{\frac{2\mathbb{V}_{t_\tau}^*(s,a)\mathbb{C}\log(SAT/\delta)}{\tau}} + \frac{\mathbb{C}7\log(SAT/\delta)}{3(1-\gamma)\tau} \\
&= \sqrt{\frac{2\mathbb{V}_{t_\tau}^*(s,a)\mathbb{C}\log(SAT/\delta)}{N_{t_\tau}(s,a)}} + \frac{\mathbb{C}7\log(SAT/\delta)}{3(1-\gamma)N_{t_\tau}(s,a)}.
\end{aligned}
\tag{D.6}
$$

In addition, for $\tau = 0$, we have

$$
\left[(\mathbb{P} - \mathbb{P}_{t_\tau+1})V^*\right](s,a) \leq \frac{1}{1-\gamma} \leq \frac{\mathbb{C}7\log(SAT/\delta)}{3(1-\gamma)\big(N_{t_\tau}(s,a) \vee 1\big)},
\tag{D.7}
$$

where the first inequality holds due to $0 \leq V^*(s) \leq 1/(1-\gamma)$ and the second inequality holds due to $N_{t_\tau}(s,a) = 0$. Since $\mathbb{P}_t, \mathbb{V}_{t-1}^*$ and $N_{t-1}(s,a)$ changed only when $t = t_\tau + 1$, we complete the proof by combining (D.6) and (D.7). $\qquad\square$

### D.4  Proof of Lemma B.7

*Proof of Lemma B.7.* For simplicity, we denote $H = \lfloor 1/(1-\gamma) \rfloor + 1, T' = \lfloor T/H \rfloor + 1$ and filtration $\mathcal{F}_t$ contained all random variables before first $t + H$ steps. Then for every $t \in [T]$, we have

$$
\begin{aligned}
\frac{1}{(1-\gamma)^2} &\geq \mathbb{E}\bigg[\Big(\sum_{i=0}^\infty \gamma^i r(s_{t+i}, a_{t+i})\Big) - V_t^\pi(s_t)|\mathcal{F}_{t-H}\bigg]^2 \\
&= \mathbb{E}\bigg[\sum_{i=0}^\infty \gamma^i\big(r(s_{t+i}, a_{t+i}) + \gamma V_{t+i+1}^\pi(s_{t+i+1}) - V_{t+i}^\pi(s_{t+i})\big)|\mathcal{F}_{t-H}\bigg]^2 \\
&= \mathbb{E}\bigg[\sum_{i=0}^\infty \gamma^{2i}\big[r(s_{t+i}, a_{t+i}) + \gamma V_{t+i+1}^\pi(s_{t+i+1}) - V_{t+i}^\pi(s_{t+i})\big]^2|\mathcal{F}_{t-H}\bigg] \\
&= \mathbb{E}\bigg[\sum_{i=0}^\infty \gamma^{2i+2}\mathbb{V}_{t+i}^\pi(s_{t+i}, a_{t+i})|\mathcal{F}_{t-H}\bigg] \\
&\geq \mathbb{E}\bigg[\underbrace{\sum_{i=0}^H \gamma^{2i+2}\mathbb{V}_{t+i}^\pi(s_{t+i}, a_{t+i})}_{X_t}|\mathcal{F}_{t-H}\bigg],
\end{aligned}
\tag{D.8}
$$

where the first inequality holds due to $0 \leq r(s,a) \leq 1, 0 \leq V_t^\pi(s) \leq 1/(1-\gamma)$ and the second inequality holds due to $\mathbb{V}_{t+i}^\pi(s_{t+i}, a_{t+i}) \geq 0$. For the random variable $X_t$, we have

$$
|X_t| \leq \sum_{i=0}^H \frac{\gamma^{2i+2}}{(1-\gamma)^2} \leq \frac{1}{(1-\gamma)^3}, \ \mathrm{Var}\big[|X_t||\mathcal{F}_{t-H}\big] \leq (\max|X_t|)\mathbb{E}[X_t|\mathcal{F}_{t-H}] \leq \frac{1}{(1-\gamma)^5},
$$

Since $X_t$ is $\mathcal{F}_t$-measurable and $\mathbb{E}[X_t|\mathcal{F}_{t-H}] \leq 1/(1-\gamma)^2$, for each $i \in [H]$, by Lemma B.2, with probability at least $1 - \delta$, we have

$$
\begin{aligned}
\sum_{j=0}^{T'} X_{jH+i} &\leq \sum_{j=0}^{T'} \mathbb{E}[X_{jH+i}|\mathcal{F}_{(j-1)H+i}] + \sqrt{\frac{2T'\log(1/\delta)}{(1-\gamma)^5}} + \frac{2\log(1/\delta)}{3(1-\gamma)^3} \\
&\leq \frac{T'}{(1-\gamma)^2} + \sqrt{\frac{2T'\log(1/\delta)}{(1-\gamma)^5}} + \frac{2\log(1/\delta)}{3(1-\gamma)^3}.
\end{aligned}
\tag{D.9}
$$

Taking summation for (D.9) with all $i \in [H]$, with probability at least $1 - H\delta$, we have

$$
\sum_{t=1}^T X_t = \sum_{i=1}^H \sum_{j=0}^{T'} X_{jH+i}
$$

$$\leq \sum_{i=1}^{H} \left( \frac{T'}{(1-\gamma)^2} + \sqrt{\frac{2T' \log(1/\delta)}{(1-\gamma)^5}} + \frac{2 \log(1/\delta)}{3(1-\gamma)^3} \right)$$

$$\leq \frac{T}{(1-\gamma)^2} + \sqrt{\frac{4T \log(1/\delta)}{(1-\gamma)^6}} + \frac{4 \log(1/\delta)}{3(1-\gamma)^4}$$

$$\leq \frac{2T}{(1-\gamma)^2} + \frac{7 \log(1/\delta)}{3(1-\gamma)^4}, \tag{D.10}$$

where the first inequality holds due to (D.9), the second inequality holds due to $T' = \lfloor T/H \rfloor + 1$ and the third inequality holds due to $x^2 + y^2 \geq 2xy$. By the definition of $X_t$, we have

$$\sum_{t=1}^{T} X_t = \sum_{t=1}^{T} \sum_{i=0}^{H} \gamma^{2i+2} \mathbb{V}_{t+i}^{\pi}(s_{t+i}, a_{t+i})$$

$$\geq \sum_{t=1}^{T} \mathbb{V}_t^{\pi}(s_t, a_t) \sum_{i=0}^{\min\{H, t-1\}} \gamma^{2i+2}$$

$$= \sum_{i=0}^{H} \gamma^{2i+2} \sum_{t=1}^{T} \mathbb{V}_t^{\pi}(s_t, a_t) - \sum_{t=1}^{H} \mathbb{V}_t^{\pi}(s_t, a_t) \sum_{i=t}^{H} \gamma^{2i+2}$$

$$\geq \frac{\gamma^2 - \gamma^{2H+4}}{1 - \gamma^2} \sum_{t=1}^{T} \mathbb{V}_t^{\pi}(s_t, a_t) - \frac{1}{(1-\gamma)^2} \sum_{t=1}^{H} \sum_{i=t}^{H} \gamma^{2i+2}, \tag{D.11}$$

where the first inequality holds due to $\mathbb{V}_t^{\pi}(s_t, a_t) \geq 0$ and the second inequality holds due to $\mathbb{V}_t^{\pi}(s_t, a_t) \leq 1/(1-\gamma)^2$. To further bound (D.11), we have

$$\frac{\gamma^2 - \gamma^{2H+4}}{1 - \gamma^2} = \frac{\gamma^2}{1 - \gamma^2}(1 - \gamma^{2H+2}) \geq \frac{\gamma^2}{1 - \gamma^2}(1 - \gamma^{2/(1-\gamma)}) \geq \frac{4 \cdot \gamma^2}{5(1 - \gamma^2)} \geq \frac{2\gamma^2}{5(1-\gamma)}, \tag{D.12}$$

where the first inequality holds since $2H + 2 = 2\lfloor 1/(1-\gamma) \rfloor + 2 \geq 2/(1-\gamma)$, the second inequality holds since $0 \leq \gamma^{1/(1-\gamma)} \leq 0.4$ when $0 \leq \gamma \leq 1$, the last one holds since $1 + \gamma \leq 2$. We also have

$$\sum_{t=1}^{H} \sum_{i=t}^{H} \gamma^{2i+2} \leq \sum_{t=1}^{H} \frac{\gamma^{2t+2}}{1 - \gamma^2} \leq \frac{\gamma^4}{(1-\gamma^2)^2} \leq \frac{\gamma^4}{(1-\gamma)^2}. \tag{D.13}$$

Substituting (D.12) and (D.13) into (D.11), we have

$$\sum_{t=1}^{T} X_t \geq \frac{2\gamma^2}{5(1-\gamma)} \sum_{t=1}^{T} \mathbb{V}_t^{\pi}(s_t, a_t) - \frac{\gamma^4}{(1-\gamma)^4}. \tag{D.14}$$

Finally, substituting (D.14) into (D.10), we have

$$\gamma^2 \sum_{t=1}^{T} \mathbb{V}_t^{\pi}(s_t, a_t) \leq \frac{5T}{1-\gamma} + \frac{35 \log(1/\delta)}{6(1-\gamma)^3} + \frac{5\gamma^4}{2(1-\gamma)^3} \leq \frac{5T}{1-\gamma} + \frac{25 \log(1/\delta)}{3(1-\gamma)^3}.$$

Thus, we complete the proof. □

### D.5 Proof of Lemma B.8

*Proof of Lemma B.8.* On the event $\mathcal{E}_7$, we have

$$\sum_{i=1}^{T} (\mathbb{V}^*(s_i, a_i) - \mathbb{V}_i^{\pi}(s_i, a_i)) \leq \sum_{i=1}^{t} \left[ \mathbb{P}\big((V^*)^2 - (V_{i+1}^{\pi})^2\big) \right](s_i, a_i)$$

$$= \sum_{i=1}^{T} \left[ \mathbb{P}(V^* - V_{i+1}^{\pi})(V^* + V_{i+1}^{\pi}) \right](s, a)$$

$$\leq \frac{2}{1-\gamma} \sum_{i=1}^{T} \left[ \mathbb{P}(V^* - V_{i+1}^{\pi}) \right](s_i, a_i)$$

$$\leq \frac{2}{1-\gamma} \sum_{i=1}^{T} (V^*(s_{i+1}) - V_{i+1}^{\pi}(s_{i+1})) + \frac{\sqrt{2T\log(1/\delta)}}{(1-\gamma)^2}$$

$$\leq \frac{2}{1-\gamma} \text{Regret}'(T) + \frac{\sqrt{2T\log(1/\delta)}}{1-\gamma} + \frac{2}{(1-\gamma)^2},$$

where the first inequality holds because of Lemma 5.1, the second inequality holds due to $0 \leq V^*(s), V_{i+1}^{\pi}(s) \leq \frac{1}{1-\gamma}$, the third inequality holds due to the definition of $\mathcal{E}_7$ and the last inequality holds due to $0 \leq V^*(s) \leq V_i(s) \leq 1/1-\gamma$. Thus, we complete the proof. $\qquad\square$

### D.6 Proof of Lemma B.9

*Proof of Lemma B.9.*

$$\sum_{i=1}^{T} (\mathbb{V}_{i-1}(s_i, a_i) - \mathbb{V}_i^{\pi}(s_i, a_i)) = \sum_{i=1}^{T} \mathbb{E}_{s' \sim \mathbb{P}_{i-1}(\cdot|s_i, a_i)}[V_{i-1}^2(s')] - \mathbb{E}_{s' \sim \mathbb{P}_{i-1}(\cdot|s_i, a_i)}[V_{i-1}(s')]^2$$

$$- \sum_{i=1}^{T} \mathbb{E}_{s' \sim \mathbb{P}(\cdot|s_i, a_i)}[V_{i+1}^{\pi}(s')^2] - \mathbb{E}_{s' \sim \mathbb{P}(\cdot|s_i, a_i)}[V_{i+1}^{\pi}(s')]^2$$

$$\leq \underbrace{\sum_{i=1}^{T} \mathbb{E}_{s' \sim \mathbb{P}_{i-1}(\cdot|s_i, a_i)}[V_{i-1}^2(s')] - \mathbb{E}_{s' \sim \mathbb{P}(\cdot|s_i, a_i)}[V_{i-1}^2(s')]}_{I_1}$$

$$+ \underbrace{\sum_{i=1}^{T} \mathbb{E}_{s' \sim \mathbb{P}(\cdot|s_i, a_i)}[V_{i-1}^2(s')] - \mathbb{E}_{s' \sim \mathbb{P}(\cdot|s_i, a_i)}[V_{i+1}^{\pi}(s')^2]}_{I_2}$$

$$+ \underbrace{\sum_{i=1}^{T} \mathbb{E}_{s' \sim \mathbb{P}(\cdot|s_i, a_i)}[V^*(s')]^2 - \mathbb{E}_{s' \sim \mathbb{P}_{i-1}(\cdot|s_i, a_i)}[V^*(s')]^2}_{I_3},$$

where the inequality holds due to $V_{i-1}(s') \geq V^*(s') \geq V_{i+1}^{\pi}(s')$.

By the definition of event $\mathcal{E}_8$, we have

$$\left\| \mathbb{P}_{i-1}(\cdot|s, a) - \mathbb{P}(\cdot|s, a) \right\|_1 \leq \frac{\sqrt{2S\mathbb{C}\log(T/\delta)}}{\sqrt{N_{i-1}(s, a) \vee 1}}. \tag{D.15}$$

Thus, for the term $I_1$, since $0 \leq V_{i-1}^2(s') \leq 1/(1-\gamma)^2$, we have

$$I_1 \leq \sum_{i=1}^{T} \frac{\sqrt{2S\mathbb{C}\log(T/\delta)}}{(1-\gamma)^2 \sqrt{N_{i-1}(s_i, a_i) \vee 1}} \leq \frac{S\sqrt{2AT\mathbb{C}\log(3T)\mathbb{C}\log(T/\delta)}}{(1-\gamma)^2}, \tag{D.16}$$

where the first inequality holds due to (D.15) and the second inequality holds due to Lemma B.3. For the term $I_2$, on the event $\mathcal{E}_6$, we have

$$I_2 \leq \sum_{i=1}^{T} \left[ \mathbb{P}\big( (V_{i-1})^2 - (V_{i+1}^{\pi})^2 \big) \right](s_i, a_i)$$

$$= \sum_{i=1}^{T} \left[ \mathbb{P}(V_{i-1} - V_{i+1}^{\pi})(V_{i-1} + V_{i+1}^{\pi}) \right](s, a)$$

$$\leq \frac{2}{1-\gamma} \sum_{i=1}^{T} \left[ \mathbb{P}(V_{i-1} - V_{i+1}^{\pi}) \right](s_i, a_i)$$

$$\leq \frac{2}{1-\gamma}\sum_{i=1}^{T}(V_{i-1}(s_{i+1}) - V_{i+1}^{\pi}(s_{i+1})) + \frac{\sqrt{2T\mathbb{C}\log(2/\delta)}}{1-\gamma}$$

$$\leq \frac{4S}{1-\gamma} + \frac{2}{1-\gamma}\sum_{i=1}^{T}(V_{i+1}(s_{i+1}) - V_{i+1}^{\pi}(s_{i+1})) + \frac{\sqrt{2\mathbb{C}\log(T/\delta)}}{(1-\gamma)^2}$$

$$\leq \frac{2}{1-\gamma}\text{Regret}'(T) + \frac{\sqrt{2T\mathbb{C}\log(1/\delta)}}{(1-\gamma)^2} + \frac{4S+2}{(1-\gamma)^2}, \tag{D.17}$$

where the first inequality holds due to $V_{i-1}(s') \geq V^*(s') \geq V_{i+1}^{\pi}(s')$, the second inequality holds due to $0 \leq V_{i-1}(s'), V_{i+1}^{\pi}(s') \leq 1/(1-\gamma)$, the third inequality holds due to the definition of event $\mathcal{E}_6$ and the forth inequality holds due to $V_{i-1}(s') \geq V_{i+1}(s')$.

For the term $I_3$, since $0 \leq V^*(s')^2 \leq 1/(1-\gamma)^2$, on the event $\mathcal{E}_8$, we have

$$I_3 \leq \sum_{i=1}^{T}\frac{\sqrt{2S\log(T/\delta)}}{(1-\gamma)^2\sqrt{N_{i-1}(s_i,a_i)}} \leq \frac{S\sqrt{2AT\log(T/\delta)\log(3T)}}{(1-\gamma)^2}, \tag{D.18}$$

where the first inequality holds due to (D.15) and the second inequality holds due to Lemma B.3. Taking an union bound for (D.16), (D.17) and (D.18), with probability at least $1 - 3\delta$, we have

$$\sum_{i=1}^{t}(\mathbb{V}_{i-1}(s_i,a_i) - \mathbb{V}_i^{\pi}(s_i,a_i)) \leq \frac{2\text{Regret}'(T)}{1-\gamma} + \frac{9S\sqrt{2AT\log(T/\delta)\log(3T)}}{(1-\gamma)^2}.$$

$\square$

## D.7 Proof of Lemma B.10

*Proof of Lemma B.10.* For each $i \in [H]$, $s \in \mathcal{S}$ and $t \in [T]$, if $N_t(s) = 0$, the we have

$$\text{Regret}'(t,s,h) = 0 \leq \frac{16SAU^2\sqrt{N_t(s)}}{(1-\gamma)^{2.5}} + \frac{20S^2A^{1.5}U^{4.5}}{(1-\gamma)^{3.5}}.$$

Otherwise, we have

$$\text{Regret}'(t,s,h) = \sum_{1\leq i\leq t, s_i=s}\gamma^h\big[V_{i+h}(s_{i+h}) - V_{i+h}^{\pi}(s_{i+h})\big]$$

$$= \sum_{1\leq i\leq t, s_i=s}\gamma^h\big[Q_t(s_{i+h},a_{i+h}) - V_{i+h}^{\pi}(s_{i+h})\big]$$

$$\leq \sum_{1\leq i\leq t, s_i=s}\gamma^{h+1}[\mathbb{P}_{i+h-1}V_{i+h-1}](s_{i+h},a_{i+h}) + \gamma^h\text{UCB}_{i+h-1}(s_{i+h},a_{i+h})$$

$$- \gamma^{h+1}\mathbb{P}V_{i+h+1}^{\pi}(s_{i+h},a_{i+h})$$

$$= I_1 + I_2 + I_3 + \gamma^h I_4 + \text{Regret}'(t,s,h+1), \tag{D.19}$$

where the first inequality holds due to definition update rule (4.2). $I_1,\ldots,I_4$ are defined as follows.

$$I_1 = \sum_{1\leq i\leq t, s_i=s}\gamma^{h+1}(V_{i+h-1}(s_{i+h+1}) - V_{i+h+1}(s_{i+h+1})),$$

$$I_2 = \sum_{1\leq i\leq t, s_i=s}\gamma^{h+1}[(\mathbb{P}_{i+h-1} - \mathbb{P})V_{i+h-1}](s_{i+h},a_{i+h}),$$

$$I_3 = \sum_{1\leq i\leq t, s_i=s}\gamma^{h+1}\big[\mathbb{P}(V_{i+h-1} - V_{i+h+1}^{\pi})\big](s_{i+h},a_{i+h}),$$

$$- \gamma^{h+1}\big[V_{i+h-1}(s_{i+h+1}) - V_{i+h+1}^{\pi}(s_{i+h+1})\big],$$

$$I_4 = \sum_{1\leq i\leq t, s_i=s}\text{UCB}_{i+h-1}(s_{i+h},a_{i+h}).$$

For the term $I_1$, we have

$$\sum_{1 \le i \le t, s_i = s} \gamma^{h+1}(V_{i+h-1}(s_{i+h+1}) - V_{i+h+1}(s_{i+h+1})) \le \sum_{i=1}^{t} \sum_{s' \in \mathcal{S}} V_{i+h-1}(s') - V_{i+h+1}(s')$$

$$\le \frac{2S}{1 - \gamma}, \tag{D.20}$$

where the first inequality holds due to $V_{i+h-1}(s') \ge V_{i+h+1}(s')$ and the second inequality holds due to $0 \le V_t(s) \le 1/(1 - \gamma)$.

For the term $I_2$, with probability at least $1 - \delta$, we have

$$\sum_{1 \le i \le t, s_i = s} \gamma^{h+1} [(\mathbb{P}_{i+h-1} - \mathbb{P}) V_{i+h-1}] (s_{i+h}, a_{i+h})$$

$$\le \sum_{1 \le i \le t, s_i = s} \frac{\gamma^{h+1} \sqrt{2SU}}{(1 - \gamma) \sqrt{N_{i+h-1}(s_{i+h}, a_{i+h}) \vee 1}}$$

$$\le \frac{\gamma^{h+1} \sqrt{2SU}}{(1 - \gamma)} \sqrt{N_t(s) \sum_{1 \le i \le t, s_i = s} \frac{1}{N_{i+h-1}(s_{i+h}, a_{i+h}) \vee 1}}$$

$$\le \frac{\sqrt{2SU}}{1 - \gamma} \sqrt{N_t(s) SAU}$$

$$= \frac{SU \sqrt{2N_t(s)A}}{1 - \gamma}, \tag{D.21}$$

where the first inequality holds due to Lemma B.1 and the definition of $U$, the second inequality holds due to Cauchy-Schwarz inequality and the third inequality holds due to Lemma B.3.

For the term $I_3$, Since the random process $s_{i+h+1} \sim \mathbb{P}(\cdot | s_{i+h}, a_{i+h})$ is dependent with whether $s_{i+1}, .., s_{i+h+1} = s$, we cannot directly use Lemma B.1 to bound this term. However, we can use the same technique in the proof of Lemme B.7, which divide the time horizon into $H$ sub-horizon and use Lemma B.1 for each sub-horizon. Compared with the upper bound of $I_3$ in proof of Theorem 4.5, this technique will lead to a gap of $\sqrt{H}$ and we have

$$\sum_{i \le t, s_i = s} \gamma^{h+1} [\mathbb{P}(V_{i+h-1} - V_{i+h+1}^\pi)] (s_{i+h}, a_{i+h}) - \gamma^{h+1} [V_{i+h-1}(s_{i+h+1}) - V_{i+h+1}^\pi(s_{i+h+1})]$$

$$\le \frac{\sqrt{2N_t(s)U}}{(1 - \gamma)} \sqrt{H}$$

$$\le \frac{2U \sqrt{N_t(s)}}{(1 - \gamma)^{1.5}}, \tag{D.22}$$

where the second inequality holds due to the definition of $U$. For the term $I_4$, we have

$$\sum_{1 \le i \le t, s_i = s} \text{UCB}_{i+h-1}(s_{i+h}, a_{i+h})$$

$$\le \underbrace{\sum_{1 \le i \le t, s_i = s} \sqrt{\frac{8U \mathbb{V}_{i+h-1}(s_{i+h}, a_{i+h})}{N_{i+h-1}(s_{i+h}, a_{i+h}) \vee 1}}}_{I_{41}} + \underbrace{\sum_{1 \le i \le t, s_i = s} \frac{8U}{(1 - \gamma)(N_{i+h-1}(s_{i+h}, a_{i+h}) \vee 1)}}_{I_{42}}$$

$$+ \underbrace{\sum_{1 \le i \le t, s_i = s} \sqrt{\frac{8 \sum_{s'} \mathbb{P}_{i+h}(s' | s_{i+h}, a_{i+h}) \min\{100 B_{i+h}(s'), 1/(1 - \gamma)^2\}}{N_{i+h-1}(s_{i+h}, a_{i+h}) \vee 1}}}_{I_{43}}. \tag{D.23}$$

For the term $I_{41}$, with probability at least $1 - \delta$, we have

$$\sum_{1 \le i \le t, s_i = s} \sqrt{\frac{8U \mathbb{V}_{i+h-1}(s_{i+h}, a_{i+h})}{N_{i+h-1}(s_{i+h}, a_{i+h}) \vee 1}}$$

$$\leq \sqrt{8U}\sqrt{\sum_{1\leq i\leq t,s_i=s}\mathbb{V}_{i+h-1}(s_{i+h},a_{i+h})}\sqrt{\sum_{1\leq i\leq t,s_i=s}\frac{1}{N_{i+h-1}(s_{i+h},a_{i+h})\vee 1}}$$

$$\leq U\sqrt{8SA}\sqrt{\sum_{1\leq i\leq t,s_i=s}\mathbb{V}_{i+h-1}(s_{i+h},a_{i+h})}$$

$$\leq U\sqrt{8SA}\sqrt{\frac{2N_t(s)}{(1-\gamma)^2}}, \tag{D.24}$$

where the first inequality holds due to Cauchy-Schwarz inequality, the second inequality holds due to Lemma B.3, the last inequality holds due to $0\leq \mathbb{V}_{i+h-1}(s_{i+h},a_{i+h})\leq 1/(1-\gamma)^2$.

For the term $I_{42}$, by Lemma B.3, we have

$$\sum_{1\leq i\leq t,s_i=s}\frac{8U}{(1-\gamma)\big(N_{i+h-1}(s_{i+h},a_{i+h})\vee 1\big)}\leq \frac{8SAU^2}{1-\gamma}. \tag{D.25}$$

For the term $I_{43}$, with probability at least $1-2\delta$, we have

$$\sum_{1\leq i\leq t,s_i=s}\sqrt{\frac{8\sum_{s'}\mathbb{P}_{i+h}(s'|s_{i+h},a_{i+h})\min\big\{100B_{i+h}(s'),1/(1-\gamma)^2\big\}}{N_{i+h-1}(s_{i+h},a_{i+h})\vee 1}}$$

$$\leq \sqrt{8\sum_{1\leq i\leq t,s_i=s}\frac{1}{N_{i+h-1}(s_{i+h},a_{i+h})\vee 1}}$$

$$\cdot\sqrt{\sum_{1\leq i\leq t,s_i=s}\sum_{s'}\mathbb{P}_{i+h}(s'|s_{i+h},a_{i+h})\min\Big\{100B_{i+h}(s'),\frac{1}{(1-\gamma)^2}\Big\}}$$

$$\leq \sqrt{8SAU}\sqrt{\sum_{1\leq i\leq t,s_i=s}\sum_{s'}\mathbb{P}_{i+h}(s'|s_{i+h},a_{i+h})\min\Big\{100B_{i+h}(s'),\frac{1}{(1-\gamma)^2}\Big\}}$$

$$\leq \sqrt{8SAU}\bigg[\sum_{1\leq i\leq t,s_i=s}\bigg(\frac{\sqrt{SU}}{(1-\gamma)^2\sqrt{N_{i+h-1}(s_{i+h},a_{i+h})\vee 1}}$$

$$+\sum_{s'}\mathbb{P}(s'|s,a)\min\Big\{100B_{i+h}(s'),\frac{1}{(1-\gamma)^2}\Big\}\bigg)\bigg]^{1/2}$$

$$\leq \sqrt{8SAU}\bigg[\frac{SU\sqrt{AN_t(s)}}{(1-\gamma)^2}+\frac{\sqrt{2N_t(s)U}}{(1-\gamma)^2}$$

$$+\sum_{1\leq i\leq t,s_i=s}\min\Big\{\frac{100S^2A^2U^5}{(1-\gamma)^5\big(N_{i+h-1}(s_{i+h+1})\vee 1\big)},\frac{1}{(1-\gamma)^2}\Big\}\bigg]^{1/2}$$

$$\leq \sqrt{8SAU}\sqrt{\frac{SU\sqrt{AN_t(s)}}{(1-\gamma)^2}+\frac{\sqrt{2N_t(s)U}}{(1-\gamma)^2}+\frac{100S^3A^2U^6}{(1-\gamma)^5}}, \tag{D.26}$$

where the first inequality holds due to Cauchy-Schwarz inequality, the second inequality holds due to Lemma B.3, the third inequality holds due to Lemma B.1, the forth inequality holds due to Lemma B.1 and the last inequality holds due to Lemma B.3. Substituting (D.20), (D.21), (D.22), (D.23) into (D.19), with probability at least $1-4H\delta$, we have

$$\text{Regret}'(t,s,h)\leq \text{Regret}'(t,s,h+1)+\frac{16SAU\sqrt{N_t(s)}}{(1-\gamma)^{1.5}}+\frac{20S^2A^{1.5}U^{3.5}}{(1-\gamma)^{2.5}}. \tag{D.27}$$

Notice that

$$\text{Regret}'(t,s,H)=\sum_{1\leq i\leq t,s_i=s}\gamma^H\big[V_{i+H}(s_{i+H})-V^\pi_{i+H}(s_{i+H})\big]$$

$$\leq \sum_{1 \leq i \leq t, s_i = s} \frac{\gamma^H}{1 - \gamma}$$

$$\leq \sum_{1 \leq i \leq t, s_i = s} \frac{1}{T}$$

$$\leq 1,$$

where the first inequality holds due to $V_{i+H}(s_{i+H}) - V_{i+H}^\pi(s_{i+H}) \leq 1/(1-\gamma)$ and the second inequality holds due to definition of $H$. Thus, taking summation of (D.27) with all $h \in [H]$, with probability at least $1 - H^2\delta$, we have

$$\text{Regret}'(t, s, 0) \leq \frac{16SAU^2\sqrt{N_t(s)}}{(1-\gamma)^{2.5}} + \frac{20S^2A^{1.5}U^{4.5}}{(1-\gamma)^{3.5}}. \tag{D.28}$$

In addition, if $N_t(s) > 0$, we have

$$V_t(s) - V^*(s) \leq \frac{1}{N_t(s)} \sum_{1 \leq i \leq t, s_i = s} V_i(s) - V^*(s)$$

$$\leq \frac{1}{N_t(s)} \sum_{1 \leq i \leq t, s_i = s} [V_i(s) - V_i^\pi(s)]$$

$$\leq \frac{16SAU^2}{(1-\gamma)^{2.5}\sqrt{N_t(s)}} + \frac{20S^2A^{1.5}U^{4.5}}{(1-\gamma)^{3.5}N_t(s)},$$

where the first inequality holds due to $V_i(s)$ is decreasing, the second inequality holds due to $V^*(s) \geq V_i^\pi(s)$ and the third inequality holds due to (D.28). Notice that when $N_t(s) \geq S^2AU^3/(1-\gamma)^2$, we have

$$V_t(s) - V^*(s) \leq \frac{16SAU^2}{(1-\gamma)^{2.5}\sqrt{N_t(s)}} + \frac{20S^2A^{1.5}U^{4.5}}{(1-\gamma)^{3.5}N_t(s)} \leq \frac{36SAU^2}{(1-\gamma)^{2.5}\sqrt{N_t(s)}}.$$

Otherwise, we have

$$V_t(s) - V^*(s) \leq \frac{1}{1-\gamma} \leq \frac{36SAU^2}{(1-\gamma)^{2.5}\sqrt{N_t(s)}}.$$

Thus, we complete the proof of Lemma B.10. $\qquad\square$