# OpenReview forum: "Nearly Minimax Optimal Reinforcement Learning for Discounted MDPs"
_NeurIPS.cc/2021/Conference — NeurIPS 2021 Poster_

### Official Review · Reviewer_2f6X · 2021-07-01

**Rating:** 7
**Confidence:** 3

**Summary:**

The authors study the problem of learning a discounted tabular-MDP. They adopt a notion of regret that is closely related to policy identification problem.

The authors make two main contributions:
1. They show how to adapt UCBVI to discounted MDP. This lead to an algorithm called UCBVI-\gamma algorithm. This model-based algorithm uses OFU principle by computing Bernstein-type bonus for empirical Q-function. The policy of the algorithm is updated every time step (online algorithm).  The author prove that its regret is upper bounded by \tilde{O}(\sqrt{SAT}/(1-\gamma)^1.5) where \gamma is the discount factor.

2. The authors also derive a minimax lower bound for the problem that essentially match the upper bound of the algorithm {\Omega}(\sqrt{SAT}/(1-\gamma)^1.5). This shows that their algorithm is minimax-optimal.


**Ethical Concerns:**

NA.

**Limitations And Societal Impact:**

The authors could comment on the practical use of the algorithm (numerical complexity and performance compared to related work).

Apart from the regret proof, is there a big difference with UCB-VI? It seems to be that UCBVI-\gamma does not use episode. Is it the main difference?

The regret definition impliees that only the visited states will be well identified. Would an episodic approach with restart be more appropriate here?


**Main Review:**

In summary, I found that the main result is strong but the paper is a bit incremental. I find the result strong because compared to previous work, the paper closes the gap between the lower and upper bound for discounted MDPs (more precisely, the the order of the two bounds provided are the same up to logarithmic factors and [huge] constants). That being said, the algorithm to obtain the upper bound is essentially a variant of UCBVI. I found the counter-example for the lower bound more original.

The definition of the regret seems adapted for disconted MDP. It is the sum over time steps of the differences between the optimal expected return of an optimal policy and the expected return of the computed policy, both at current states. As indicated it is closely related to policy identification but only for those states. I am wondering if it make sense as those states might not be representative of the whole MDP. Would it make sense to restart the MDP from time to time (for instance after a geometrically distributed time?).

Also, there is no numerical simulations comparing the algorithm with other existing algorithms and no detail about algorithm's computational complexity.  My guess is that it is probably because, in practice, the proposed algorithm is not efficient: despite having order-optimal regret, the hidden constant in the O(.) are just huge. Hence, this algorithm has mostly a theoretical interest but not really a practical interest.

The sketches of proof for both theorems are convincing. The exposition of the paper is clear and the obtained result is nearly minimax optimal for tabular discounted MDPs. The literature review is well described.


**Time Spent Reviewing:**

3

---

> ### Author Response · Authors · 2021-08-10
> **Response to Reviewer 2f6X**
>
> Thank you for your detailed comments.
>
> ---
> **Q1:** Apart from the regret proof, is there a big difference with UCB-VI? It seems to be that UCBI-\gamma does not use episodes. Is it the main difference? (Novelty)
>
> **A1:** The main difference is that our setting differs from previous work (Azar et al., 2017): finite horizon (episode length $H$) v.s. Discounted infinite horizon (discounted factor $\gamma$). These two settings are totally different. Besides, there are further differences in the algorithm design and proofs, as we will explain in detail as follows.
>
> First, in the algorithm design, (Azar et al., 2017) estimates the Q-value functions in a backward way, which is not feasible in the discounted setting due to the lack of ‘episode’ definition. In contrast, our algorithm estimates them in a forward way, which brings additional difficulties to analyze (because the Bellman equation is a backward equation). Second, while both analyses use the law of total variance to obtain the optimal regret, we need to construct truncated Markov chains to apply the law of total variance (see the proof of Lemma B.7).
>
> ---
> **Q2:** The regret definition implies that only the visited states will be well identified… those states might not be representative of the whole MDP
>
> **A2:** You are right that our definition of regret only cares about the states that will be visited. It has been used by quite a few recent works to evaluate the RL algorithms for discounted MDPs. We also want to clarify that it is impossible to make policy identification for unseen states or actions in both episodic MDPs and discounted MDPs unless a generative model is used or further assumption is made. Other performance measures such as the regret for episodic MDPs [3] and the sample complexity of exploration for discounted MDPs [9] also have the same issue. Therefore, we think our regret definition is a meaningful performance measure.
>
>
>
> ---
> **Q3:**  Would an episodic approach with restart be more appropriate here?
>
> **A3:** Your suggestion to convert a discounted MDPs into episodic MDPs by restarting is interesting. However, we feel that directly learning discounted MDPs (without restart) is more natural.
>
>
> ---
> **Q4:** Simulation (performance compared to related work)
>
> **A4:** Thank you for your suggestion. We have run the numerical experiments to compare our UCBVI-$\gamma$ with several baseline algorithms. We run these algorithms on the hard-to-learn MDPs proposed in Theorem 4.4 with $S = 10$, $A=4$, $\gamma = 0.9$, $\epsilon = 0.1$.
> We compare our algorithm UCBVI-$\gamma$ with three baselines:
> 1. $\epsilon$-greedy Q-Learning ($\epsilon$-greedy for short): $\epsilon = 0.1$
> 2. Q-Learning with UCB: Dong et al., 2019
> 3. Double Q-learning: Liu and Su (2020)
>
> In the experiment, we make a simulation with T=4000 (repeating 100 times and take the average) and compare the reward with the optimal policy (always choose the best action). Here is the result of the regret.
>
> | Algorithm \Steps |   500    |  1000  |  1500 |     2000   | 2500       | 3000      | 3500     | 4000     |
> |:-------------:    |:---------: |:--------:|:--------:|:---------:   |:------:       |:--------:   |:---------: |:---------:  |
> |$\epsilon$-greedy       |      54.1     |  105.3    | 148.1    | 186.6    | 231.4 | 276.9  | 316.1   |354.2 |
> |Q-Learning with UCB   |    56.7       | 103.0   |  143.8     |173.8     |203.9  |231.8  | 246.7  | 257.9|
> |Double Q-learning       |    57.1       |  101.5    |  143.6  |  181.0   |  218.6  | 253.7 | 280.3  | 298.8|
> |UCBVI-$\gamma$      |    52.5      |   89.1    |    121.4   |  145.5  |  178.1  | 210.1 | 229.7  | 246.8|
>
> From the experiment results, we can see that our algorithm can achieve lower regret than all the other algorithms.
>
> ---
> **Q5:** Computation cost
>
> **A5:** In the $UCBI-\gamma$ algorithm, we update the value function $Q_t, V_t$ with value iteration, and it will cost $O(S^2A)$ for each update, which leads to an $O(S^2AT)$ computation cost in total. Such a computation cost can be further reduced by considering the ‘batch’ update scheme adapted in (Jaksch et al. 2010; Dann & Brunskill 2015), which only updates the value function whenever the number of visits $N_t(s, a)$ doubles. With this method, the number of updates can be upper bounded by $O(SA\log T)$ and the total cost for updating the value function is $(S^3A^2 \log T)$. In addition, choosing the action with respect to the value function will cost $O(AT)$ time complexity. We will provide a detailed discussion in the final version.

---

> > ### Comment · Reviewer_2f6X · 2021-08-31
> > **Clear answer**
> >
> > I would like to thank the authors for their detailed answer. It clarified most of my points. I increased my score (6->7).

---

> > > ### Author Response · Authors · 2021-08-31
> > > **Thank you!**
> > >
> > > Thank you for your positive feedback and for increasing the score. We will prepare the final version by incorporating all the revisions following your comments and suggestions.

---

### Official Review · Reviewer_bvo4 · 2021-07-16

**Rating:** 7
**Confidence:** 3

**Summary:**

This paper closes the gap betewwn upper and lower bounds for regret of tabular discounted MDPs. In particular, it proposes UCBVI-$\gamma$ algorithm that achieves regret of $\widetilde{O}\left(\frac{\sqrt{SAT}}{(1-\gamma)^{1.5}}\right)$ and constructs a class of hard MDPs such that any algorithm will have expected regret at least $\widetilde{\Omega}\left(\frac{\sqrt{SAT}}{(1-\gamma)^{1.5}}\right)$.

**Limitations And Societal Impact:**

The authors adequately addressed the limitation of their work in Section 6 by pointing out the requirement on $T$ for their algorithm to be nearly optimal. The potential negative social impact is not addressed, which is not so needed for a purely theoretical analysis on tabular MDPs.

**Main Review:**

Overall, although the techniques used do not seem to be very novel, I consider the results of this paper as a significant progress on tabular disconted MDPs. Meanwhile, the writing of this paper is also clear and easy to follow.

Questions:
- On page 9, how you show that $$\mathbb{E}\left[\sum_{i=K+1}^{T_j}V^*(X_{j, i})-\frac{r(X_{j, i}, A_{j, i})}{1-\gamma}\mid T_j>K\right]\geq -T,$$ or more specifically, $V^*(X_{j, i})-\frac{r(X_{j, i}, A_{j, i})}{1-\gamma}\geq -1$?

---

Good paper. Score remains the same after rebuttal.

**Time Spent Reviewing:**

6

---

> ### Author Response · Authors · 2021-08-10
> **Response to Reviewer bvo4**
>
> Thank you for your positive comments.
>
> ---
> **Q1:** On page 9, why $V*(X_{j,i})-r(X_{j,i},A_{j,i})/(1-\gamma)\ge -1$?
>
> **A1:** Thanks for pointing out this typo. You are right that the lower bound of this term should be $-1/(1-\gamma)$, which implies that the lower bound of $E[\sum_{i=K+1}^{T_j}V*(X_{j,i})-r(X_{j,i},A_{j,i})/(1-\gamma)| T_j>K]$ should be $-T/(1-\gamma)$. However, we want to clarify that this term is only used to control the error in the small probability event (probability is smaller than $\delta$), and $\delta$ only appears in logarithmic factors in other terms. Therefore, our proof can still go through by replacing the current $\delta$ with a smaller $\delta’$, which only differs from $\delta$ by some logarithmic factors. We will fix this typo in the final version.

---

> > ### Comment · Reviewer_bvo4 · 2021-08-16
> > **Follow-up Question**
> >
> > Thank you very much for your response and my question is well-addressed. However, I want to add a follow-up question about the definition of regret, which is raised by reviewer 2f6X.
> >
> > I can see that your definition of regret is also used in other works. What I'm concerning is that this definition of regret sometimes does not reflect a good policy that we desire. As an extreme example, suppose there is an absorbing state with zero reward that is connected to the initial state $s_1$ and can be reached deterministically by some action $a'$. Then, a policy that takes $a'$ initially will incur regret at most only $\frac{1}{1-\gamma}$. However, this is usually a policy that we should avoid because it gives absolutely no reward even though the other policy can potentially incur larger regret. That is, we get a policy that we want to avoid by minimizing the regret. On the other hand, this is not a problem for finite-horizon MDP.
> >
> > I still appreciate the work you have done under your definition of regret, but I want to hear how you think about this phenomenon and suggest adding a discussion section about this in the paper.

---

> > > ### Author Response · Authors · 2021-08-16
> > > **Re:Follow-up Question**
> > >
> > > Thank you for your positive feedback and insightful question. Your comment on the regret and the example you raised are indeed correct. Nevertheless, we would like to clarify that the issue that you pointed out is not a limitation of the regret definition, but a fundamental issue for online learning of discounted MDPs. Besides the regret used in our paper, there is another widely used performance measure called sample complexity of exploration, which is defined as the number of suboptimality gaps that are greater than $\epsilon$ (See, e.g., Lattimore and Hutter, 2012, Dong et al., 2019). It can be shown that the issue you pointed out is unavoidable no matter using the regret or sample complexity of exploration as the performance measure, since there is no “restarting” scheme like that in episodic MDPs to prevent the agent from being stuck in those bad states. We will be sure to add a detailed discussion on this fundamental limitation about online learning of discounted MDPs in the final version.

---

### Official Review · Reviewer_3A7m · 2021-07-16

**Rating:** 6
**Confidence:** 3

**Summary:**

This paper proposes a new model-based algorithm for tabular MDPs, and proves a upper bound and a lower bound on regret.

**Limitations And Societal Impact:**

This paper focuses on theoretical analysis. It's expected that the paper does not have direct negative societal impact.

**Main Review:**

Pros:
The newly proposed algorithm enriches the collection of model-based algorithms. The power of $T$ in the upper bound on regret is decreased to 0.5 by this paper.

Cons:
In Section 6, it says "There is still a gap between the upper and lower bounds when $T\leq\max\{S^3A/(1-\gamma)^4,SA/(1-\gamma)\}$." But Theorem 4.2 and Theorem 4.4 respectively require $T\geq S^3A^2/(1-\gamma)^4$ and $T\geq SA/(1-\gamma)^4$. Hence only when $T\geq S^3A^2/(1-\gamma)^4$, the upper bound matches the lower bound. In this case, where does $T\leq\max\{S^3A/(1-\gamma)^4,SA/(1-\gamma)\}$ in Section 6 come from? Furthermore, how restrictive is this condition on $T$? There seems no discussion about this in the paper.

I've read the authors' response and updated my score accordingly.

**Time Spent Reviewing:**

3

---

> ### Author Response · Authors · 2021-08-10
> **Response to Reviewer 3A7m**
>
> Thank you for your insightful comments.
>
> ---
> **Q1:** Some typos in Section 6 ($T\leq \max\{S^3A/(1-\gamma)^4, SA/(1-\gamma)\}$)
>
> **A1:** We apologize for the typos in Section 6 and it should be $T\leq \max\{S^3A^2/(1-\gamma)^4, SA/(1-\gamma)^4\}$. We will fix them in the final version.
>
> ---
> **Q2:** How restrictive is this condition on T
>
> **A2:** The condition on $T$ is not very restrictive. Since $S, A,\gamma$ are all fixed parameters for the underlying MDP, the condition can be satisfied when $T$ is large enough. This basically requires the agent to interact with the environment sufficiently. The only reason that we impose such a condition is that we want to make both upper and lower bounds match with each other to achieve minimax optimality, which is similar to many existing works (e.g., [3, 4]). We will explore how to remove this condition in future work.

---

### Official Review · Reviewer_QtQ3 · 2021-07-16

**Rating:** 7
**Confidence:** 3

**Summary:**

This work studies the problem of learning stochastic discounted infinite-horizon Markov Decision Processes (MDPs) with unknown transition. The authors propose an algorithm UCBVI-gamma based on recent advances in episodic MDPs (e.g. Azar et al., 2017; Jin et al., 2018; Zanette and Brunskill, 2019) and specifically the Bernstein-type exploration bonus. Besides, the authors provide a lower bound which indicates that UCBVI-gamma matches the minimax lower bound up to logarithmic factors.

**Limitations And Societal Impact:**

This work is pure theoretical and does not have any potential negative societal impact.

**Main Review:**

1. Contribution

In this work, the authors propose the model-based algorithm UCBVI-gamma which achieves O(sqrt(SAT)/ (1-gamma)^1.5) regret, which matches the lower bound on the expected regret O(sqrt(SAT)/(1-gamma)^1.5). This is the first algorithm that matches the lower bound. The algorithm design is inspired from UCBVI-BF of Azar et al. (2017), which adopts a specific Bernstein-type exploration bonus.

On the other hand, although based on the hard two-state instance of Liu and Su (2020), the proof of the lower bound in Theorem 4.4 is novel and inspiring.

2. Weaknesses

The proving techniques and algorithm design of the upper bound are quite similar to that of previous works such as Azar et al. (2017) for UCBVI-BF. I am willing to reconsider my score if the authors could emphasize their own technical contributions in the proof of the upper bound.

Besides, the third term in Theorem 4.2 and the decomposition of Eq (5.1) would benefit from more high-level explanations.

3. Writing Issues

line 262-336, the format of reference seems incorrect. It should not be numbered.


**Time Spent Reviewing:**

17

---

> ### Author Response · Authors · 2021-08-10
> **Response to Reviewer QtQ3**
>
> Thank you for your encouraging comments!
>
> ---
> **Q1:** Our technical contribution
>
> **A1:**  First, our setting differs from previous work (Azar et al., 2017): finite horizon v.s. Discounted infinite horizon. Second, in the algorithm design, (Azar et al., 2017) compute the Q-value functions in a backward way, which is not feasible in the discounted setting due to the lack of the ‘episode’ concept. In contrast, our algorithm constructs them in a forward way, which brings additional difficulties to analyze. Third, while both analyses use the law of total variance to obtain the optimal regret, we need to construct truncated Markov chains to apply the law of total variance (see the proof of Lemma B.7), which we believe is a new and interesting technical contribution.
>
> ---
> **Q2:**  Explanation of the third term in Theorem 4.2 and the decomposition of Eq (5.1)
>
> **A2:** The third term in Theorem 4.2 comes from the randomness in the stochastic transition process (term $I_5$ in Eq (5.1)), and it can be further bounded by Azuma-Hoeffding inequality (line 197). For the decomposition of Eq (5.1), terms $I_2$ and $I_3$ measure the estimation error of matrix P, term $I_1$ controls the estimation error from the value function $V_{t-1}$, term $I_4$ comes from the exploration bonus term in the Algorithm and term $I_5$ accounts for the randomness in the stochastic transition process. We will add more explanation in the revision.
>
> ---
> **Q3:** The format of reference
>
> **A3:** Thanks for your suggestion. We will revise it in the revision.

---

### Decision · Program_Chairs · 2021-09-27

**Decision:**

Accept (Poster)

**Comment:**

This paper studies the sample complexity of tabular (finite state/action) reinforcement learning in the infinite-horizon discounted setting, which is a fundamental problem in reinforcement learning theory. The main contribution is to provide improved upper and lower bounds on the optimal sample complexity (w.r.t, the dependence on the horizon parameter (1-gamma)^(-1)) which resolve the optimal sample complexity. The reviewers found these results to be interesting and technically non-trivial and recommend acceptance, but the authors are encouraged to take their suggestions into account and update the paper to better emphasize what the key analysis ideas and why they are novel.